

# Internal tide signatures on surface chlorophyll concentration in the Brazilian Equatorial Margin

Carina Regina de Macedo[1,2,3], Ariane Koch-Larrouy[3,4], José Carlos Bastos da Silva[5,6], Jorge Manuel Magalhães[5,7], Fernand Assene[3,8], Manh Duy Tran[2], Isabelle Dadou[3], Amine M'Hamdi[3,4,9], Trung Kien Tran[2], and Vincent Vantrepotte[2]

[1]Earth Observation and Geoinformatics Division, National Institute for Space Research (INPE), São José dos Campos, Brazil.
[2]Univ. Littoral Côte d'Opale, CNRS, Univ. Lille, IRD, UMR 8187 - LOG - Laboratoire d'Océanologie et de Géosciences, F-62930 Wimereux, France.
[3]LEGOS CNRS/IRD/CNES, Université de Toulouse, Toulouse, France.
[4]CECI CNRS/Cerfacs/IRD, Université de Toulouse, Toulouse, France.
[5]Department of Geosciences, Environment and Spatial Planning, Faculdade de Ciências da Universidade do Porto, Rua do Campo Alegre 687, 4169-007, Porto, Portugal.
[6]Instituto de Ciências da Terra, Polo Porto, Universidade do Porto, Rua do Campo Alegre 687, 4169-007, Porto, Portugal.
[7]CIIMAR, Universidade do Porto, Rua dos Bragas 289, 4050-123, Porto, Portugal.
[8]Department of Maritime Navigation and Information Systems, National Advanced School of Maritime and Ocean Science and Technology (ENSTMO), University of Ebolowa, BP: 292 Kribi, Cameroun.
[9]Departamento de Oceanografia da Universidade Federal de Pernambuco – DOCEAN/UFPE, Recife, Brazil.

**Correspondence:** Carina Regina de Macedo (macedo.nina@gmail.com)

**Abstract.** This study investigates the influence of tides on chlorophyll-a (CHL) variability in the Brazilian Equatorial Margin using daily remotely sensed CHL data from 2005 to 2021. The impact of the tides is deduced by comparing the spring tides with the neap tide signal (fortnightly signal, 14.7 days) in the GlobColour and MODIS-AQUA spring-neap tide composites. Results show that, on the shallow Amazon shelf, significant fortnightly CHL variability is primarily driven by barotropic
tide-induced friction on the shelf that produces a significant vertical mixing. On the northwestern shelf, where the Amazon River plume dominates, sediment resuspension (likely driven by stronger tidal mixing during spring tides) suppresses primary production. In contrast, the increased stratification of the river plume during neap tides enhances nutrient availability likely enhancing primary production, leading to negative spring-neap CHL differences (GlobColour: -50%, MODIS-AQUA: -84%). Conversely, the northeastern shelf, where low turbid waters are present, exhibits positive CHL differences (GlobColour: +30%,
MODIS-AQUA: +70%), likely caused by nutrient-rich uplift. Offshore, baroclinic tides, also known as internal tides (ITs), induce CHL-positive spring-neap tide differences in the spring-neap tide composite with a spatial structure of a wave-like pattern along IT pathways. These anomalies are spaced by mode-2 wavelengths (about 68 km), with peak values reaching +3.3% (GlobColour) and +9.0% (MODIS-AQUA). The observed wave-like pattern may be attributed to two potential mechanisms. First, tide aliasing caused by the satellites' orbit characteristics, which consistently capture IT wave crests at similar loca-
tions, with concurrent modulation of the deep chlorophyll maximum (DCM) due to IT wave passage along the thermocline. Second, the propagation of internal tides (ITs) as beams into the ocean interior ("Ray theory"), causing upward displacement of isotherms and driving nutrient fluxes that enhance primary production, supported by ray-tracing analysis. Wave patterns





in CHL spring-neap tide composites from GlobColour and MODIS-AQUA suggest contributions from mode-1 and mode-2
internal tides. Results indicate also a lag of 1-3 days between spring-neap tides and peak chlorophyll variability, indicative of
maximum mixing. The effects of ITs on CHL are more pronounced than on sea surface temperature, likely due to differences
in sensor penetration depths and the influence of air-sea interactions.

## 1 Introduction

The Brazilian Equatorial Margin (BEM) is characterized by intense activity of internal tides (ITs) (Tchilibou et al., 2022),
also known as baroclinic tides. The seasonal variability in water stratification, currents, and mesoscale circulation significantly
influences IT activity in this region. From March to July, the pycnocline is shallower, slightly stronger, and more horizontally
homogeneous, along with weaker currents and reduced mesoscale activity (Richardson and Walsh, 1986; Richardson et al.,
1994; Silva et al., 2005; Aguedjou et al., 2019; Tchilibou et al., 2022; Assene, 2024). Between August and December, the
pycnocline becomes deeper and slightly weaker, the North Equatorial Countercurrent (NECC) intensifies, and the eddy kinetic
energy (EKE) increases, leading to longer IT wavelengths (Barbot et al., 2021; Tchilibou et al., 2022). The BEM is also known
by intense activity of internal solitary waves (ISWs). ISWs observed off the Amazon shelf originate from the disintegration
of ITs occurring several hundred kilometers from their generation sites along the steep Amazon shelf break (Magalhães et al.,
2016). The water stratification, currents and mesoscale circulation from August to December further modulate the ISW charac-
teristics, leading to a greater variability in terms of ISW diversity and its mean wavelengths (Magalhães et al., 2016; de Macedo
et al., 2023). Furthermore, de Macedo et al. (2023) found that ISW activity in the region is greater during spring tides than
neap tides.

Baroclinic tides are generated when barotropic tidal currents interact with steep, variable bathymetry in a stratified ocean
(Munk and Wunsch, 1998; Vlasenko et al., 2005; Gerkema and Zimmerman, 2008). Both barotropic and baroclinic tides
facilitate the transfer of energy from larger to smaller scales in the ocean through a tidal energy cascade. Strong ITs off the
Amazon shelf can induce vertical mixing by dissipating part of their energy locally at the shelf break, as well as along their
propagation path, with maximum dissipation occurring approximately every 120 km, corresponding to the mode-1 wavelength
(Tchilibou et al., 2022; Assene, 2024). Offshore mixing along the IT propagation path can cause sea surface temperature (SST)
to cool up to 0.3°C with a higher cooling in the thermocline up to 1.2°C (Assene et al., 2024). Mixing induced by barotropic
and baroclinic tides can significantly affect water mass formation and properties, influencing primary production and water
temperature (da Silva et al., 2002; Koch-Larrouy et al., 2007; Hu et al., 2008; Sharples, 2008; Muacho et al., 2014; González-
Haro et al., 2019; Kossack et al., 2023; Assene et al., 2024; Jacobsen et al., 2023; Capuano et al., 2025; M'hamdi et al.,
2025).





In shallow coastal waters, barotropic tides primarily dissipate their energy through friction with the seabed, generating bottom-driven mixing that alters water properties. This mixing can enhance primary production by replenishing nutrients in the surface mixed layer (Hu et al., 2008; Sharples, 2008). However, this shelf mixing can also inhibit phytoplankton growth by promoting resuspended sediments, which reduces the photosynthetically available radiation (PAR) (Cloern, 1991; Kossack et al., 2023).

As interfacial ITs propagate, they cause vertical displacements in the pycnocline (of some tens of meters, see e.g., Fig. 1 in da Silva et al. (2002)) and hence displace passive phytoplankton cells within the water column. This movement shifts the depth of the chlorophyll maximum (DCM) either above or below the light penetration depth, resulting in, respectively, increased or decreased chlorophyll-a (hereinafter referred to as CHL) concentrations detected by remote sensors (da Silva et al., 2002; Muacho et al., 2014; Kim et al., 2018; M'hamdi et al., 2025). The vertical displacement of phytoplankton by ITs can alter the light available for primary production (Lande and Yentsch, 1988; Jacobsen et al., 2023). ITs also enhance the vertical mixing of nutrients into the DCM, thus supporting primary production (Sharples et al., 2007; Tuerena et al., 2019; Kossack et al., 2023; Jacobsen et al., 2023). M'hamdi et al. (2025) demonstrated using a Slocum G2 glider deployed off the Amazon shelf during AMAZOMIX 2021 cruise that the enhancement of the CHL concentration associated with the passage of ITs may be due to the two combined effects: 1) the ITs modulate the DCM, causing its vertical displacement and oscillation as it rises and deepens in response to IT propagation. This movement may enhance phytoplankton's light exposure, stimulating primary production; 2) Mixing events associated with ITs increase CHL concentration in both the surface and bottom layers of the water column. Jacobsen et al. (2023) studied the response of primary production to IT beams based on model simulations configured for an oligotrophic system with a nutricline depth below 50 meters. They found that while subsurface light limitation is reduced for passive plankton within tidal beams, leading to higher primary production rates, the dominant effect of tidal beams on primary production is the increased nutrient supply to the euphotic zone near tidal beam generation locations.

Tides are characterized by spring-neap tide cycles (fortnightly cycles, approximately 14.675 days), which impact water mass properties. According to Sharples (2008), the fortnightly modulation of tidal mixing affects primary production in stratified shelf areas; changes in the timing of the spring-neap cycle could account for up to 10% of the inter-annual variability in bloom timing in regions with weak tidal regimes, and up to 25% in areas with stronger tidal currents. In the Indonesian Sea, fortnightly variations in chlorophyll (CHL) concentration in shallow coastal waters have been attributed to the spring-neap cycle of barotropic ocean currents (Zaron et al., 2023; Capuano et al., 2025). In deeper waters, the authors hypothesized that the fortnightly CHL variability is linked to the modulation of vertical nutrient fluxes by baroclinic tidal mixing. Furthermore, the baroclinic energy flux, dependent on baroclinic currents along the IT pathways, is twice as strong during the spring tide compared to the neap tide in the Brazilian Equatorial Margin (Tchilibou et al., 2022; Assene, 2024). Sharples et al. (2007) observed that in a stratified water column of 200 meters depth at the shelf edge of the Celtic Sea, the upward mixing of nutrients across the thermocline, driven by an internal tide, resulted in a much higher nitrate flux into the surface layer during the spring tide than during the neap tide, thereby fueling new primary production. In the northwestern Amazon shelf, Assene (2024) found maximum values of fortnightly temperature variability ( 0.15°C). The author showed that the fortnightly modulation of temperature is enhanced in the thermocline with a horizontal structure similar to the baroclinic Sea Surface Height (SSH) along





IT pathways. The temperature mixing and vertical advection due to the ITs play the main contribution on the temperature vari-
ability in the thermocline. They also noted a 2-3 days delay between the astronomical tidal forcing and the peak of temperature
variability.

This paper investigates the impact of barotropic and baroclinic tides on the CHL concentration and SST in the Brazilian
Equatorial Margin. It explores, for the first time, the fortnightly signal in remote sensing measurements of surface CHL con-
centration in this region, also accounting for various delays related to astronomical tidal forcing. The paper is structured as
follows: Section 2 outlines the data and methods, while Section 3 presents results from two showcase studies, wavelet analy-
sis, and spring-neap tide composites. Section 4 discusses the influence of ITs on CHL concentration based on these findings.
Finally, Section 5 summarizes the key conclusions.

## 2   Data and Methods

### 2.1   Remote sensing data processing

ISW signatures were identified in two showcase images acquired on September 28th, 2007, and October 12th, 2018, over the
Amazon shelf in the sunglint region. These images consist of Level 1B data from the Moderate Resolution Imaging Spectrora-
diometer (MODIS) sensor onboard the TERRA satellite, using band 6 (centered at 1640 nm) with a spatial resolution of 500 m.
The MODIS-Terra images were obtained from NASA's Earth Science Data System (ESDS) (https://doi.org/10.5067/MODIS/MOD
02QKM.NRT.061).

CHL information were derived from daily acquisitions of Level 1A data from the MODIS onboard the AQUA satellite,
covering the period from January 1st, 2005, to December 31st, 2021. The Level 1A MODIS/AQUA images were obtained from
NASA's Ocean Color website (https://oceancolor.gsfc.nasa.gov/) and processed to Level 1C using the SeaDAS 8.1.0 software
package (https://seadas.gsfc.nasa.gov/). Remote sensing reflectance (Rrs) was computed from the Level 1C MODIS data using
POLYMER 4.13 (https://www.hygeos.com/polymer) for atmospheric correction. POLYMER is specifically designed to recover
marine reflectance and is particularly effective in correcting for sunglint effects (Steinmetz et al., 2011). Rrs values were
then used to estimate near-surface CHL concentrations through NASA's standard algorithm implemented in SeaDAS. This
algorithm combines an updated version of the OC3M band ratio algorithm (O'Reilly and Werdell, 2019) with the color index
(CI) approach (Hu et al., 2019).

Since the study area is highly affected by clouds throughout the year, we have used as well the daily L4 CHL data from
the global multi-sensor Copernicus-GlobColour processor, which is a blend between OC5 (Gohin et al., 2002) and CI of Hu
et al. (2012). GlobColour product was acquired from 1st January 2005 to 31st December 2021 in The Copernicus Marine
Service (DOI. 10.48670/moi-00281). This product is a CHL daily composite obtained by merging multiple ocean satellite
sensor acquisitions (SeaWiFS, MODIS, MERIS, VIIRS-SNPP, JPSS1, OLCI-S3A, S3B) and by applying temporal averaging
and interpolation methods to fill the missing data values.

Models based on band ratios in the visible spectrum offer reliable chlorophyll estimates for clear to moderately turbid waters
but show limitations in highly turbid environments (Tran et al., 2023). Taking heed of the presence of very turbid waters in





our study area associated with the Amazon River plume, we used the methodology developed by Tran et al. (2023) to exclude from our analysis pixels that are classified as turbid waters (optical water types 4 and 5) more than 20% in our time series. The method developed by the authors clusters the reflectance data focusing on the shape of the spectra (Vantrepotte et al., 2012; Mélin and Vantrepotte, 2015).

Gap-free maps of daily L4 foundation sea surface temperature (SST) data from the OSTIA were acquired from 01 January 2007 to 31 December 2021 in the Copernicus Marine Service (Good et al., 2020; Donlon et al., 2012; Stark et al., 2007) (https://doi.org/10.48670/moi-00165). This product uses in-situ data and merged satellite observations (AMSR2-GCOM-W, AVHRR-MetOp-B, SEVIRI-MSG, VIIRS-SNPP, NOAA-20, SLSTR-S3A, S3B) and provides SST maps at 0.05deg. x 0.05deg. horizontal grid resolution.

## 2.2    Showcase analysis

The presence of ISW signatures in two MODIS-TERRA images acquired under sunglint conditions over the BEM is used as a proxy for IT activity. To emphasize the impact of ITs on the CHL concentration, we calculated the difference between the CHL concentration on the specific day of the showcases and the average CHL values over 15 days, centered on the showcase day:

$$CHL_{difference} = CHL_{showcase} - CHL_{average} \qquad (1)$$

where

$$CHL_{average} = \frac{1}{n} \sum_{i=d-7days}^{n=d+7days} CHL_i \qquad (2)$$

$d$ represents the showcase day, while $i$ and $n$ indicate the start and end days, respectively, of the 15-day average CHL calculation. This enables the assessment of potential changes in CHL concentrations driven by ITs, compared to the CHL 135 15-day average within the study area.

## 2.3    Wavelet analysis

Wavelet analysis is a powerful tool for analyzing ecological systems since the methodology performs a local time-scale decomposition of the signal overcoming the problem of non-stationarity in time series (Lau and Weng, 1995; Torrence and Compo, 1998). The 1D continuous wavelet spectral analysis was carried out on daily CHL GlobColour and OSTIA products. The con-
tinuous Morlet wavelet transform was applied to the CHL and SST anomaly time series, which was performed using Python 3.8 and the free Python package PyCWT (https://pypi.org/project/pycwt/) based on the method developed in Torrence and Compo (1998). The power spectra significance test was made according to Torrence and Compo (1998). Xing et al. (2021) applied a similar wavelet approach to investigate CHL variability associated with spring-neap tidal cycles.



## 2.4 Spring-Neap Tide Composition

We developed another methodology to exploit the fortnight modulation of the tides on CHL by performing spring-neap tide composite maps. Tidal elevation at the Amazon shelf break (45.5°W, 1°N) was analyzed using the Tidal Toolbox, available within the COMODO tools developed and maintained by the SIROCCO (SImulation Réaliste de l'OCéan COtier) national service (https://sirocco.obs-mip.fr/other-tools/prepost-processing/comodo-tools/).

Let $S$ and $N$ represent the CHL data for spring and neap tide, respectively, during the time series period (Figure 1). Considering the lag between the maximum tidal potential and the maximum tidal elevation (age of the tides), when mixing is at its peak, we considered different days of delay from spring and neap tides, respectively, $S = \{S_{1+delay}, S_{2+delay}, \cdots, S_{n+delay}\}$ and $N = \{N_{1+delay}, N_{2+delay}, \cdots, N_{n+delay}\}$, where $delay = \{0, 1, 2, 3, 4, 5\}$ days. In contrast to wavelet analysis and other methods commonly used to examine the frequency content of IT-related signals, such as least-squares harmonic analysis (e.g., Zaron et al., 2023), our method also retrieves the sign of the spring-neap CHL variability. This allows us to assess whether

tidal influence contributes to increases or decreases in CHL concentration within the affected area. The composite maps can be calculated as follows:

$$f(S,N) = \frac{1}{n} \sum_{i=1}^{n} \left( \left( \frac{3}{4}\overline{S_{i+delay} \pm 1day} + \frac{1}{4}\overline{S_{i+1+delay} \pm 1day} \right) + \left( \frac{1}{4}\overline{N_{i+delay} \pm 1day} + \frac{3}{4}\overline{N_{i+1+delay} \pm 1day} \right) \right) \frac{1}{M} \quad (3)$$

Where $M = \left( \overline{S_{i+delay} \pm 1day} + \overline{S_{i+1+delay} \pm 1day} + \overline{N_{i+delay} \pm 1day} + \overline{N_{i+1+delay} \pm 1day} \right) / 4$. The term M is added to the equation to consider the relative variations in the CHL, removing the raw signal.

## 2.5 Ray tracing calculation


The "ray theory" describes linear IT propagation as ray-like structures moving through the water column at a specific angle to the horizontal. It provides a reasonable first-order estimate of IT beam positions. To assess their potential impact on CHL concentration, we estimated these positions under the assumption of gradual and continuous ocean stratification, where buoyancy frequency ($N$) varies slowly with depth. Using the IT ray-tracing method from da Silva et al. (2012), we calculated IT rays

from the Amazon shelf, with their slope relative to the horizontal ($s$) determined by $N$, the M2 tidal frequency ($\sigma$), and the Coriolis frequency ($f$):

$$s = \pm \left( \frac{\sigma^2 - f^2}{N(z)^2 - \sigma^2} \right)^{\frac{1}{2}} \quad (4)$$

IT beams typically originate from critical slopes which were assumed to be the position where the seafloor slope matches the local value of $s$. $N$ was derived from the climatological seawater salinity and temperature data, calculated using the daily Global

Ocean Ensemble Physics Reanalysis data (from 01 January 2005 to 31 December 2021 with 0.25-degree resolution) obtained from the Copernicus Marine Service (https://doi.org/10.48670/moi-00024). The data is produced from a numerical ocean model



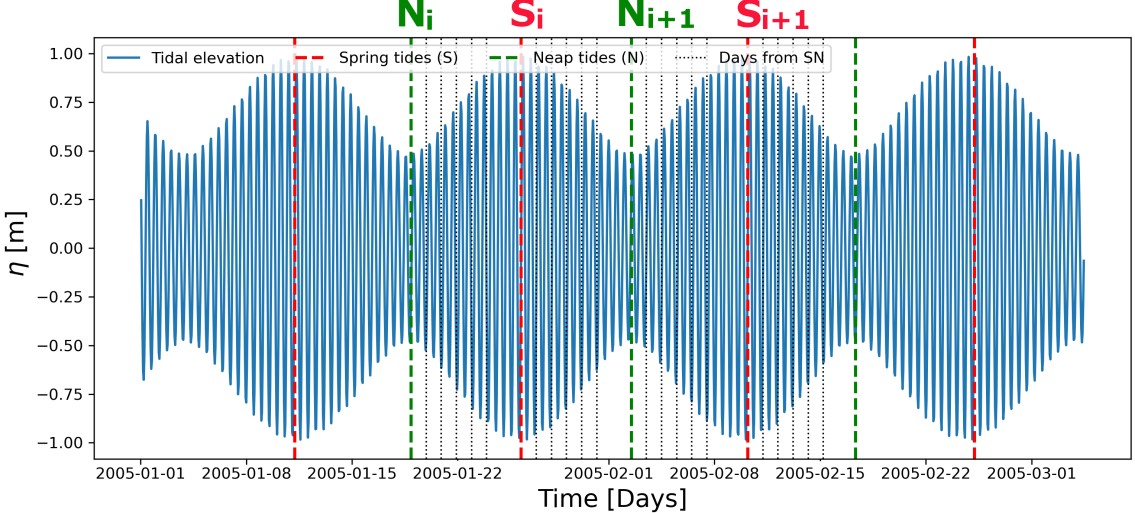

**Figure 1.** Tidal elevation analysis using the COMODO tools with the Spring-Neap modulation ( 14–15 days) at the Amazon shelf break [45.5°W, 1°N]. The red and green dashed lines indicate the times of spring and neap tides, respectively. The black dotted lines represent the days between consecutive spring and neap tides.

with data assimilation of satellite and *in situ* observations. We assume that $N$ remains constant along the horizontal pathway of IT propagation. However, in our study area, $N$ may exhibit variability due to submesoscale and mesoscale dynamics.

The bathymetry data is sourced from the 2023 GEBCO dataset, a global terrain model that offers detailed elevation informa-
tion for both ocean and land. This dataset provides elevation data at a resolution of a 15-arc-second grid (https://www.gebco.
net/data_and_products/gridded_bathymetry_data/)

## 3   Results

### 3.1   ITs showcases

We investigated daily CHL data in the Brazilian Equatorial Margin derived from MODIS-AQUA and GlobColour data for
days when ISWs signatures were identified in MODIS-TERRA images acquired under sun glint conditions (de Macedo et al.,
2023). The presence of ISWs is used as a proxy for the presence of ITs. Two significant examples (showcases) illustrating the
influence of ITs on CHL concentration were selected. In both instances, the MODIS-AQUA images were captured 2.55 hours
after the MODIS-TERRA ones, corresponding to a northeastward displacement of approximately 25 km in the ISW location.
Figures 2 and 3 present these showcases, with the leading waves of the ISW indicated by dotted-dashed black lines. The wave
positions depicted have not been adjusted for their displacement between the two remote sensing image acquisitions.

For Showcase I, shown in Figures 2-(a) and (b), daily CHL levels retrieved from MODIS-AQUA and GlobColour are
displayed, respectively. On the left side of the IT pathway A (indicated by the red dashed line), two bands of elevated CHL



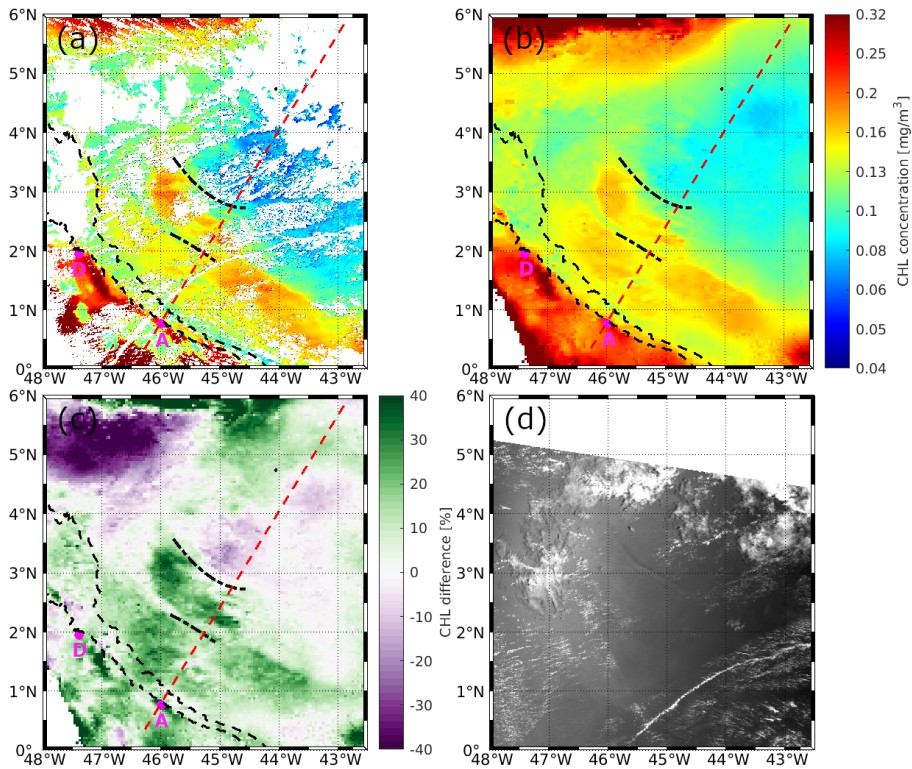

**Figure 2.** Showcase I on the influence of ITs on CHL concentration on September 28th, 2007. CHL concentration data is shown from (a) MODIS-AQUA and (b) Globcolour products. (c) CHL relative difference (in %) calculated using Equation 1. (d) ISW signatures observed in the MODIS-TERRA image. The red dashed line and magenta dots indicate the IT pathway and generation points, respectively, based on Assene et al. (2024). Black dashed lines mark the ISW signatures visible in the MODIS-TERRA image.

concentration are visible, extending parallel to the shelf break between IT generation sites A and D. The bands are separated by approximately 100 km, typical mode-1 IT wavelengths. In the location of the mapped ISW leading wave, CHL concentration

is similar to the background levels. The offshore band of enhanced CHL (around 2.5°N, 45.5°W) is sandwiched by two ISW signatures. On the right side of IT pathway A, we also observe bands of elevated CHL concentration separated by about 68 km, typical mode-2 IT wavelengths. In Showcase II (Figures 3-(a) and (b)), a similar modulation of CHL concentration is evident, with the offshore band of elevated CHL also being sandwiched by ISW signatures. The positions of the enhanced chlorophyll bands in Showcase II closely match those in Showcase I. Furthermore, although the CHL concentration is generally higher in

the GlobColours product for both showcases, the spatial patterns in both products show strong agreement.

For Showcases I and II, CHL concentration values along IT pathway A, derived from MODIS-AQUA and GlobColour data, are shown in Figures 4-(a) and (c) and Figures 4-(b) and (d), respectively. When ISW signatures cross the IT pathway, their positions are marked as vertical dashed magenta lines on the graph. Accounting for the displacement of the waves from different scene acquisitions, the recalculated wave position is indicated by a vertical dashed cyan line for the MODIS-AQUA





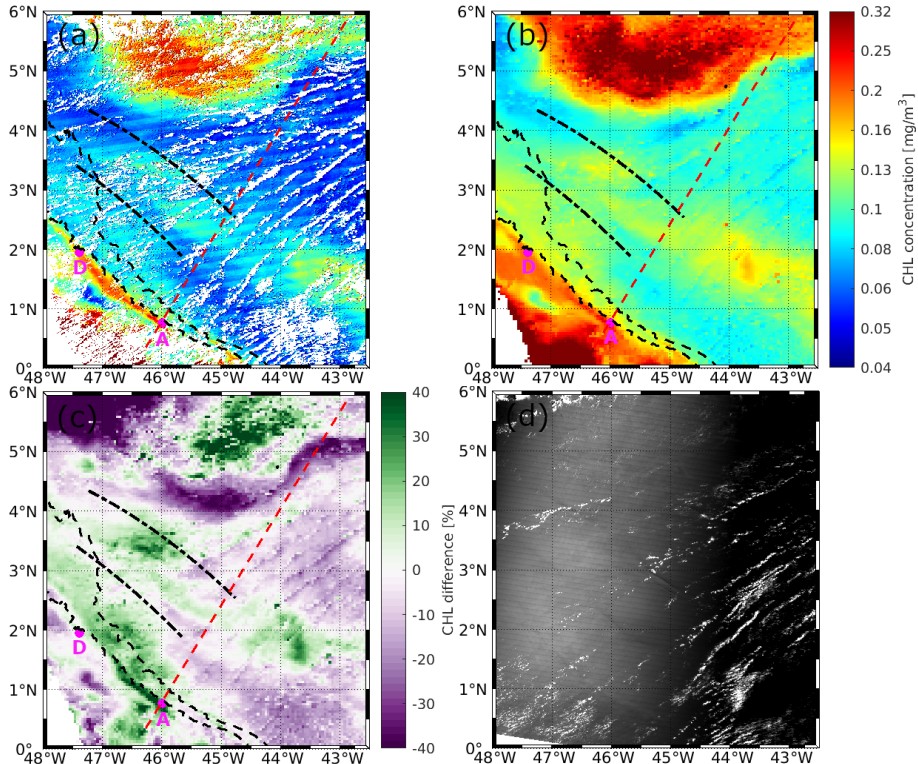

**Figure 3.** Showcase II on the influence of ITs on CHL concentration on October 12th, 2018. CHL concentration data is shown from (a) MODIS-AQUA and (b) Globcolour products. (c) CHL relative difference (in %) calculated using Equation 1. (d) ISW signatures observed in the MODIS-TERRA image. The red dashed line and magenta dots indicate the IT pathway and generation points, respectively, based on Assene et al. (2024). Black dashed lines mark the ISW signatures visible in the MODIS-TERRA image.

CHL data. The recalculated position is not plotted for GlobColour because its exact acquisition time is unknown, as it is a merged product (without a defined acquisition time). In both showcases, a distinct peak in the CHL concentration profile is observed at 262 km (202 km from the IT generation point A), with the peak being more pronounced in Showcase I when considering the MODIS-AQUA data.

CHL difference maps from GlobColour data are shown for Showcases I and II in Figures 2-(c) and 3-(c), respectively. In
both showcases, the bands of enhanced CHL correspond to concentrations higher than the 15-day average. Figures 4-(e) and (f) depict the chlorophyll relative differences along IT pathway A for both showcases. In both instances, a pronounced CHL relative difference peak of up to 45% is observed at the IT generation site A. For Showcase I, two peaks in CHL relative difference are observed at approximately 129 km and 262 km (respectively, 69 km and 202 km from IT generation point A), with the first peak reaching 19% and the second up to 31%. For Showcase II, two peaks are found at 122 km (8%) and 262 km
(7%), respectively, 62 km and 202 km from the IT generation site. A 1D horizontal band-pass filter for mode-1 and mode-2 wavelength (respectively, 100-150 km and 50-100 km, see red and blue dashed lines in Figures 4-(e) and (f)) was applied to





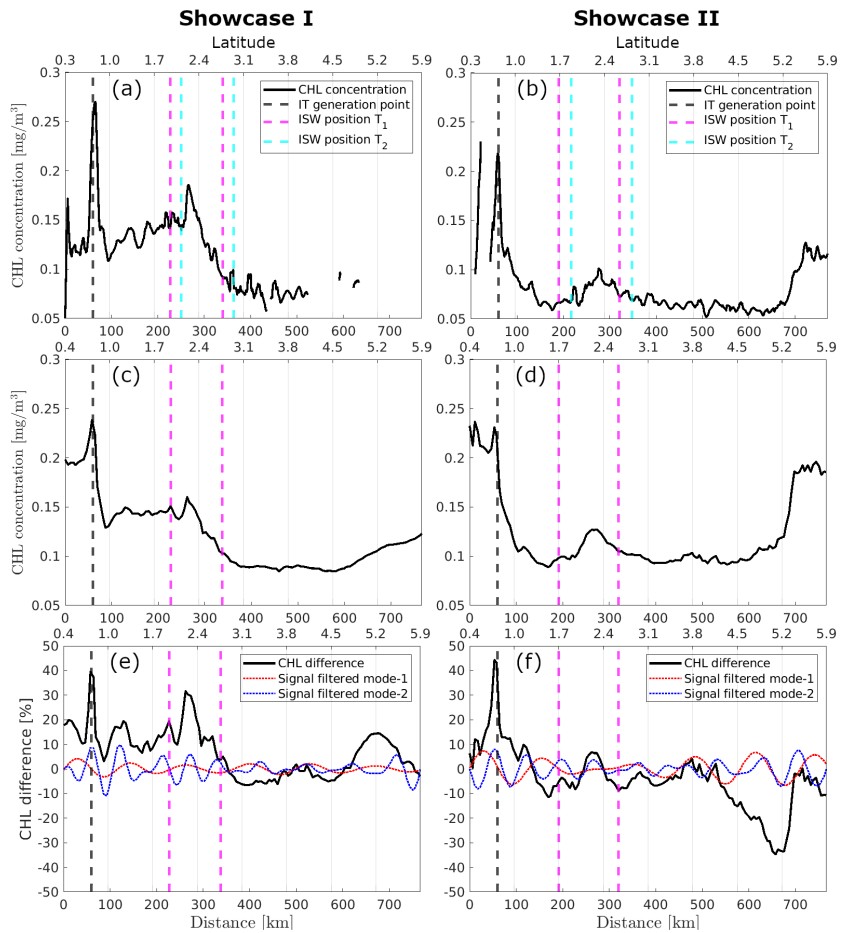

**Figure 4.** CHL concentration data along IT pathway A is shown from MODIS-AQUA for showcases (a) I and (b) II, and from the GlobColour product for showcases (c) I and (d) II. CHL relative differences calculated using Equation 1, are presented for showcases (e) I and (f) II.

highlight the impact of the IT. Modulations are stronger and align more closely with the original signal when considering the signal filtered for mode-2 wavelength in both cases, with the offshore signal weakening beyond the first 300–400 km.

### 3.2 Wavelet analysis

Figure 5-(a) shows the spatial variability of the fortnightly signal in the gap-free GlobColour CHL product, extracted using the mean Morlet wavelet power within the 14.2 to 15.2-day period where the power was statistically significantly above the 95% confidence for red noise. The SST fortnightly signal was calculated using the same methodology (see Figure 5-(b)). The depth-integrated barotropic and baroclinic energy flux (black arrows) and dissipation from Assene et al. (2024), computed using the Nucleus for European Modeling of the Ocean (NEMO v4.0.2, Madec et al., 2019) with the AMAZOMIX36 configuration, are

shown in Figures 5-(c) and (d), respectively. For both CHL and SST, the fortnightly signal is more intense in the northwest





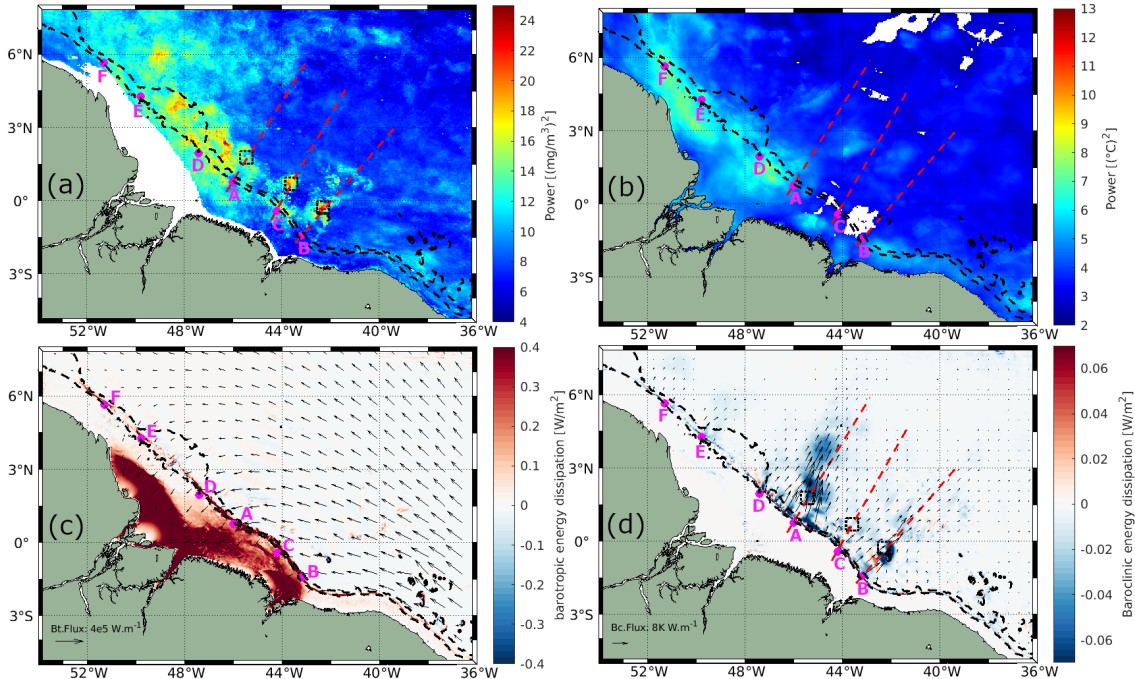

**Figure 5.** Mean Morlet wavelet power for periods between 14.2 and 15.2 days is shown for (a) CHL and (b) SST. Depth-integrated barotropic (c) and baroclinic (d) energy fluxes are represented by black arrows, with dissipation values calculated from NEMO. Black dashed rectangles indicate areas of Interest (AOIs), IT generation points identified are marked in magenta, and IT pathways are depicted by red dashed lines (Assene et al., 2024)

part of the shelf region around 46.5°W-52°W and 0°N-6°N, where barotropic tides dissipate through bottom friction, creating mixing, see Figure 5-(c). Offshore, the highest values of the CHL fortnightly signal are mostly concentrated in the northwest part while, for SST, the fortnightly signal considerably reduces. Areas with high CHL fortnightly power can also be seen aligned with the IT pathways A, B, and C (see the gray dashed rectangles in Figure 5-(a)). The areas of high power along IT

pathways A and B align well with the lower values of depth-integrated S2 baroclinic energy dissipation predicted by the model. For CHL, the strongest and weakest fortnightly signals are observed along pathways B and A, respectively.

### 3.3 Spring - Neap tide composites

Figure 6 presents the CHL spring-neap tide composites computed from Globcolour data using Equation 3, considering various delays (0 - 5 days) from spring and neap tides. Across all delays, CHL exhibits significantly higher shelf and shelf break vari-

ability than the offshore region, which is consistent with the strong shelf depth-integrated barotropic dissipation, as illustrated in Figures 5-(c). On the shelf area, a dipole pattern is evident: in the northwest region (around 47°W-52°W and 2°N-6°N), a negative anomaly in CHL is observed, with minimum values reaching approximately -50% occurring 2-3 days from the spring-




neap tide. Conversely, in the northeast region (around 40°W-47°W and 3°S-2°N), a positive anomaly is seen, with maximum values of about 30% occurring with a 1-2 day delay.

In the offshore region between IT generation points A and B, a positive spring-neap tide difference is observed, following a wave-like pattern in the horizontal structure of CHL. This wave pattern is most pronounced with a 1-day delay from the spring-neap tide, gradually diminishing in extent and coherence with longer delays. A close-up view highlighting the wave-like pattern in the CHL composite, with 1-day delay is shown in Figure A1-(a). Referring to the spring-neap tide composites with a 1-day delay as a benchmark, Figure 6-(b) illustrates at least three peaks of positive CHL spring-neap tide difference.

The profiles along the IT pathways A, B, and C for spring-neap tide composites are shown in Figure 7. Taking as reference the delay of 1 day, an average of 63 km separates the two first peaks and the second and third peaks are 83 km apart. The distances of the three peaks from IT generation points A, B, and C are detailed in Table 1. In Figure 7, the magenta stars indicate areas with frequent ISW occurrences based on findings by de Macedo et al. (2023). The third peak of positive CHL difference along pathway A is sandwiched by the regions of high ISW occurrence.

**Table 1.** Distance (km) from the IT generation sites and maximum values of CHL difference associated with the three first beams along the IT pathways A, B, and C, for GlobColour (1 day of delay from the spring-neap tides) and MODIS-AQUA data (2 days of delay).

| Data source | IT pathways | Beams | Distance (km) | CHL difference (%) |
|---|---|---|---|---|
| GlobColour | A | 1° | 128 | 2.2 |
| | | 2° | 191 | 1.3 |
| | | 3° | 291 | 1.5 |
| | B | 1° | 100 | 2.4 |
| | | 2° | 158 | 2.2 |
| | | 3° | 226 | 3.3 |
| | C | 1° | 142 | 2.3 |
| | | 2° | 211 | 2.3 |
| | | 3° | 291 | 1.3 |
| MODIS-AQUA | A | 1° | 129 | 8.4 |
| | | 2° | 182 | 9.0 |
| | | 3° | 281 | 6.6 |
| | C | 1° | - | - |
| | | 2° | 195 | 3.2 |
| | | 3° | 295 | 6.9 |





**Figure 6.** Spring-neap tide composites of CHL using daily Globcolour product, considering a delay of 0 (a), 1 (b), 2 (c), 3 (d), 4 (e), and 5 (f) days. Spring-neap tide composite map for delay of 1 (g) and 2 (h) days, using a color bar for highlighting the shelf. Areas of Interest (AOIs) are shown as black dashed rectangles, IT generation points are displayed as magenta points, IT pathways are shown as red dashed lines (Assene et al., 2024), and magenta stars represent the areas of high ISW occurrence according to de Macedo et al. (2023)



In Figure 7, focusing on IT pathway A and a 1-day delay from spring-neap tides, the positions of the first and third peaks correspond closely to the peaks of positive CHL relative difference observed in showcases I and II. Additionally, the locations of the second peak of positive spring-neap tide CHL difference along IT pathways A and C, and the third peak along IT pathway B, correspond closely with areas of high Morlet wavelet power (indicated by gray dashed rectangles in Figure 6-(b)). The peaks of CHL differences in the spring-neap tide composite, with a 1-day delay, average around 2.1% across all IT pathways. The modulation of spring-neap tide filtered for mode-1 and mode-2 IT wavelength diminishes further offshore after the initial 300-400 km. When considering the signal filtered for mode-1 and mode-2 IT wavelength, the modulations in the spring-neap tide composite map are more coherent along IT pathway C and less coherent along pathway B. However, the peaks of positive spring-neap tide CHL difference reach higher values along IT pathway B. This finding aligns with the Morlet wavelet analysis, which indicates higher fortnightly power along IT pathway B than other pathways. In contrast, lower values are observed along pathway A compared to other IT pathways. Note that, when considering a profile outside the influence of ITs, the signal filtered for mode-1 and mode-2 IT wavelengths are very low (see Figure A2 in Appendix A).

Because of the temporal averaging and interpolation methods involved in deriving CHL concentration from the GlobColour product, we also generated spring-neap tide composites using daily MODIS-AQUA imagery, as shown in Figure 8. Due to the substantial cloud cover in our study area, these composites are noisier than the GlobColour ones. Nevertheless, both datasets reproduce similar spatial patterns, particularly on the shelf region, where a comparable dipole pattern is evident in MODIS-AQUA as in GlobColour composites. In the northwest part of the shelf, MODIS-AQUA composites show maximum negative CHL differences (approximately -84%) occurring 2-3 days after spring-neap tides, whereas in the northeast region, maximum positive CHL anomalies of about 70% are observed under similar timing. Overall, CHL differences between spring and neap tides tend to be higher in MODIS-AQUA compared to GlobColour composites.

Offshore, a distinct wave pattern emerges with a 2-day delay, characterized by at least three peaks of positive CHL spring-neap tide differences along pathway A. A close-up view highlighting the wave-like pattern in the CHL composite, with 2-day delay is shown in Figure A1-(b). A similar pattern is partially visible along pathway C, which exhibits two peaks of positive CHL spring-neap tide differences. Along pathway A, the third peak of positive CHL difference is in between regions of frequent ISW occurrence, aligning with the patterns observed in GlobColour composites. Figure 9 illustrates the CHL difference profiles along the IT pathways for MODIS-AQUA composites. For a 2-day delay, along pathways A and C, the CHL spring-neap tide difference peaks average around 6.8%, more than double the signal detected in GlobColour composites. The wave-like structure begins to lose coherence after a 3-day delay from the spring-neap tide. The spatial distribution of the CHL concentration peaks closely aligns with the wave structures observed in GlobColour composites along IT pathways A and C (see Table 1). In contrast, no significant signal is detected along pathway B in the MODIS-AQUA composites. In both pathways A and C, the signal filtered for mode-2 IT wavelength is higher than the signal filtered for mode-1. The modulations in the spring-neap tide composite map are more coherent along pathway C compared to pathway A for signal filtered for mode-1 and mode-2.



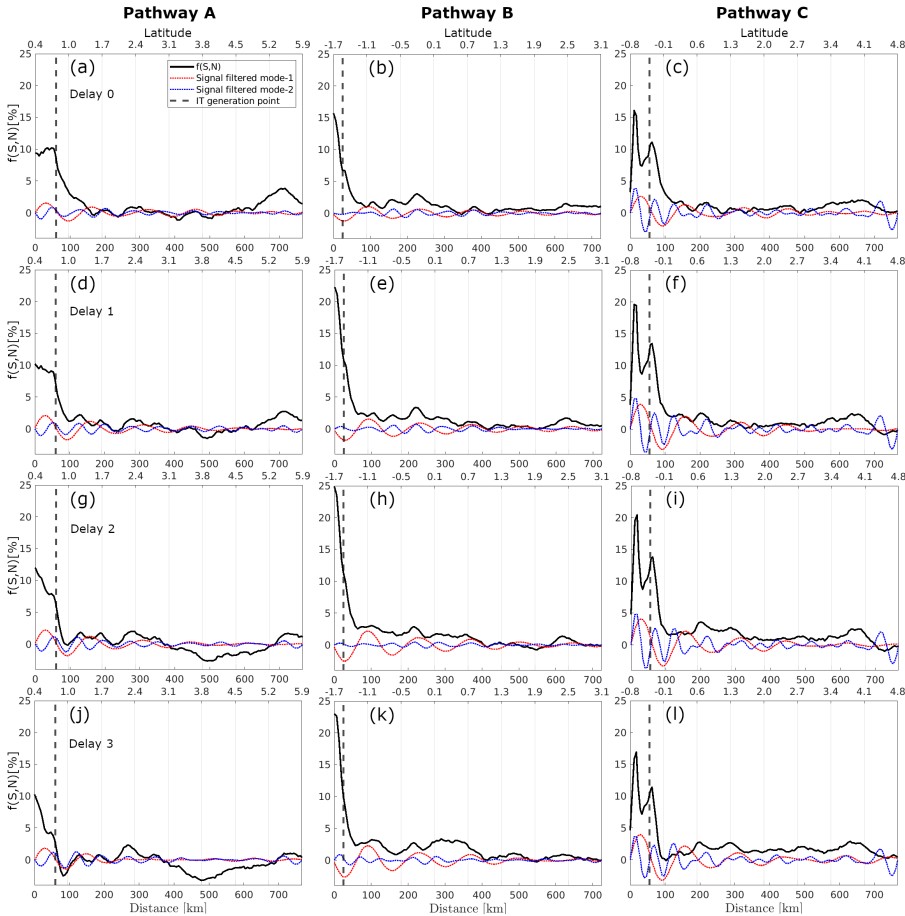

**Figure 7.** Profiles along the A, B, and C pathways (columns) of the spring-neap tide CHL composite using Globcolour, considering delays of 0-5 days. black and dashed red and blue lines represent, respectively, the original signal and the signal filtered for mode-1 (100-150 km) and mode-2 (50-100 km) wavelengths.

## 4 Discussion

### 4.1 ITs Showcases

The offshore band of enhanced CHL in showcases I and II (around 2.5°N, 45.5°W, see Figures 2 and 3) likely corresponds
to an IT crest, as two ISW signatures sandwich it. Similar bands of elevated CHL levels, sandwiched by ISW signatures,
were observed by da Silva et al. (2002) and Muacho et al. (2014) in the Bay of Biscay. They attributed the enhanced CHL
concentration to the uplifting of the DCM caused by the passage of internal tidal crests.

The CHL concentration bands associated with the IT in our showcases are 7% to 31% higher than the 15-day average,
aligning well with the findings of M'hamdi et al. (2025). Their study, based on data from a Slocum G2 glider equipped with an

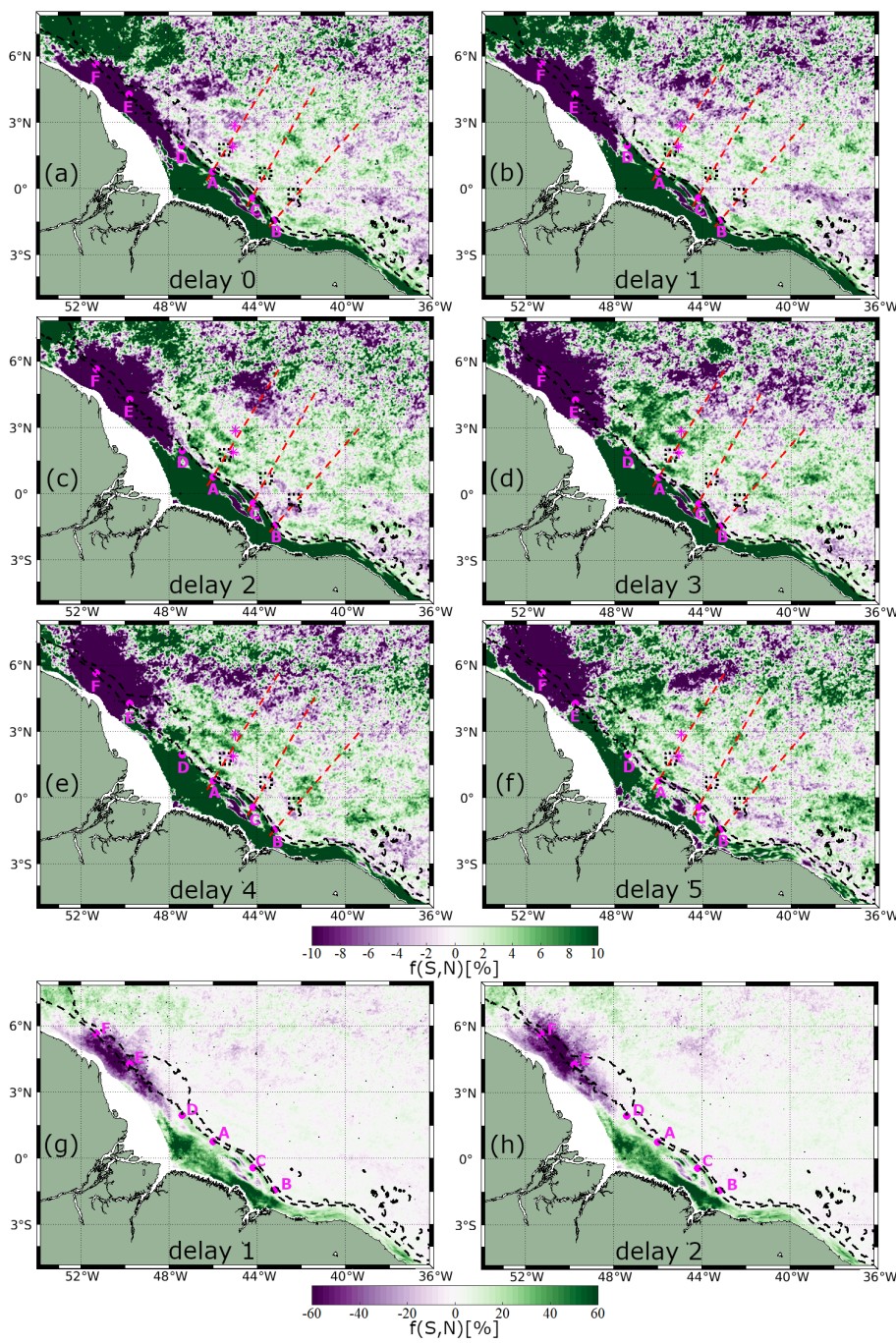

**Figure 8.** Spring-neap tide composites of CHL using daily MODIS-AQUA data, considering a delay of 0 (a), 1 (b), 2 (c), 3 (d), 4 (e), and 5 (f) days. Spring-neap tide composite map for delay of 1 (g) and 2 (h) days, using a color bar for highlighting the shelf. Areas of Interest (AOIs) are shown as black dashed rectangles, IT generation points are displayed as magenta points, IT pathways are shown as red dashed lines (Assene et al., 2024), and magenta stars represent the areas of high ISW occurrence according to de Macedo et al. (2023)



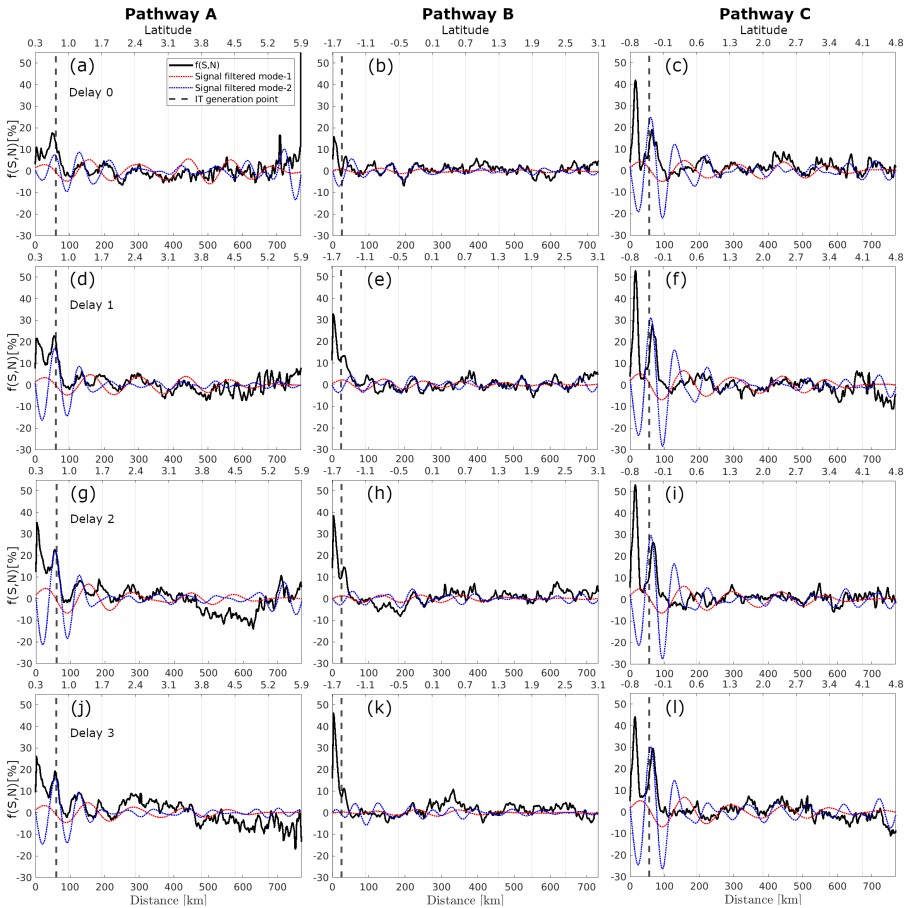

**Figure 9.** Profiles along the A, B, and C pathways (columns) of the spring-neap tide CHL composite using MODIS-AQUA, considering delays of 0-5 days. black and dashed red and blue lines represent, respectively, the original signal and the signal filtered for mode-1 (100-150 km) and mode-2 (50-100 km) wavelengths.

optical fluorescence sensor deployed off the Amazon shelf, reported that ITs increase total chlorophyll concentration by 14% to 32%.

## 4.2 Wavelet analysis

The lower fortnightly signal associated with SST compared to CHL could be attributed to two factors: 1) the OSTIA product estimates the SST at a depth of a few micrometers to millimeters in the water column (Good et al., 2020), whereas ocean
color sensors operate at wavelengths that penetrate a depth of several meters (10% of the euphotic layer). This difference in measurement depth could explain the stronger fortnightly signal seen in CHL measurements; and 2) as Assene (2024) demonstrated, ITs cool the SST and also influence upper ocean-atmosphere interactions, enhancing the net heat flux at the air-sea interface. This increased heat flux from the atmosphere to the ocean tends to damp the ITs SST cooling and to restore the





surface temperatures, resulting in a weaker fortnightly signal in SST compared to the surface CHL. Assene (2024) calculated
the sea temperature anomaly by comparing two regional simulations, one with tides and one without. They found that along IT
pathway A, surface offshore anomalies were approximately -0.2°C. These anomalies increased significantly with depth along
the IT transect, reaching up to -1.2°C at depths greater than 60 meters and up to +1.2°C at depths greater than 120 meters.

## 4.3 Spring - Neap tide composites

On the shelf, using a 1-day and 2-day delay as benchmarks for GlobColour and MODIS-AQUA CHL composites, respectively
(see Figures 6-(a) and 8-(b)), the northwest region, influenced by the turbid Amazon River plume, experiences sediment resus-
pension and tidal mixing, which may inhibit primary production due to the light limitation. In shallow, permanently mixed areas
of the northwest European shelf, tidal mixing and resuspension of suspended matter hinder primary production, as Kossack
et al. (2023) noted. According to Nittrouer et al. (2021), near the Amazon River mouth, energetic spring tides resuspend muddy
sediment in the water column, making fluid mud less dense and well-mixed but, during neap tides, the tides are less energetic
and denser fluid mud consolidates on the seabed. Another possible explanation for the negative spring-neap tide CHL anomaly
observed in this region is the influence of tidal variability on the Amazon River plume dynamics. During spring tides, strong
vertical turbulence shifts the northeastward deflection of the plume waters further offshore. In contrast, during neap tides, the
plume is deflected right upon entering the ocean remaining closer to the shore (Ruault et al., 2020). In neap tide reduced mix-
ing results in a well-stratified river plume, limiting nutrient dispersion and leading to higher nutrient concentrations within the
plume. Under these conditions, phytoplankton can access these nutrients, considering that the light availability is sufficient. In
the northeast shelf region, where lower turbid waters are present (non-plume waters), the dissipation of barotropic tides likely
promotes mixing, which may contribute to enhancing CHL concentrations. As Kossack et al. (2023) highlighted, tidal fronts
in the northwest European shelf promote vertical mixing of nutrients, enhancing primary production, with approximately 16%
of annual mean primary production attributed to tidal forcing in these regions.

### 4.3.1 Tide-aliasing effect

Figure 10 illustrates the tidal amplitudes and phases observed in all images composing the MODIS-AQUA time series. Im-
agery acquired during spring tides approximately corresponds to low tide, while acquired during neap tides approximately
corresponds to high tide. As noted by Valente and da Silva (2009), this tide-aliasing effect arises from the sun-synchronous
orbit of the MODIS-AQUA satellite relative to local tidal patterns. Consequently, it is reasonable to hypothesize that the spring-
neap tide composite signal may be influenced by the low-high tide chlorophyll dynamics. However, variations in CHL levels
between low and high tides are anticipated to be minimal, both in deep waters and coastal regions. Blauw et al. (2012) observed
a semi-diurnal pattern in the Southern North Sea, where surface phytoplankton concentrations decreased during high and low
tides, coinciding with periods of weak tidal mixing. In contrast, chlorophyll levels increased during transitional tidal phases
(flood and ebb), when current speeds and mixing intensity were stronger. In this study, flood and ebb data were excluded,
suggesting that the tide-aliasing effect is unlikely to significantly impact our results. Furthermore, it is expected that 6-hour
periodic variations in chlorophyll are more pronounced in coastal areas and considerably weaker in the open ocean.





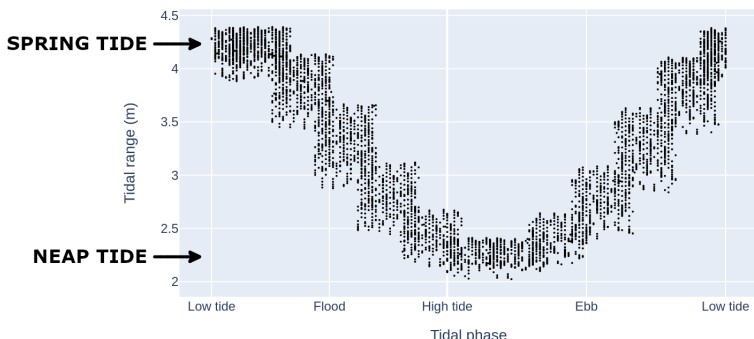

**Figure 10.** Illustration of the tidal range during the acquisition of all MODIS-AQUA scenes used in this study from 2005 to 2021.

### 4.3.2 Wave-like pattern observed in the spring-neap tide composites

The wave-like pattern observed in the spring-neap tide composites between IT generation sites A and C correlates well with the steric sea-surface height (SSSH) gradient along the northeast direction from the Hybrid Coordinate Ocean Model (HYCOM), as computed by Solano et al. (2023) (refer to their Figure 8), where SSSH modulation is observed at approximately 65-70 km intervals in that region. Given the similarity between the horizontal structures of CHL variability in the spring-neap tide composite maps and SSSH patterns, it is likely that the propagation of semi-diurnal internal tides and resulting mixing are primarily responsible for the observed positive horizontal variability in CHL spring-neap tide composites (barotropic tides are considered negligible beyond the shallow shelf, as shown in Figures 5-(c)). Furthermore, the third peak of positive CHL difference along pathway A is sandwiched by the regions of high ISW occurrence (see Figures 6 and 8), suggesting that the variation in CHL concentration in that area is likely linked to the passage of an IT crest.

The wave-like pattern could be attributed to two different effects: 1) Tidal aliasing combined with the modulation of the DCM induced by the passage of interfacial IT waves. Note that, this would mean that MODIS-AQUA consistently captures IT crests in nearly the same location since both IT generation and MODIS-AQUA orbit are synchronized with the M2 tidal constituent. Nevertheless, further investigation is required to confirm this explanation, particularly since two semidiurnal tidal cycles correspond to the merged and averaged one-day GlobColour product. 2) IT beams which potentially reduce subsurface light limitation for primary production and driving nutrient fluxes that further boost primary production (Althaus et al., 2003; Jacobsen et al., 2023; Kouogang et al., 2024). To further investigate this possibility, we applied a ray-tracing methodology in our study area, (see Figure 11). Our results reveal that the CHL peaks in the GlobColour composites align closely with regions where IT beams radiate near the surface. Considering that the positions of CHL peaks in the MODIS-AQUA data align well with those in the GlobColour composites, the alignment between CHL peak locations and the IT ray suggests that the wave patterns observed in both datasets may be driven by tidal beams, which play a role in enhancing primary production.

*Incoherent tides*

It is reasonable to assume that the wave pattern in the composite maps is likely partially averaged out by interactions of the ITs with background circulation features, including eddies, currents, and stratification. These interactions generate incoherent



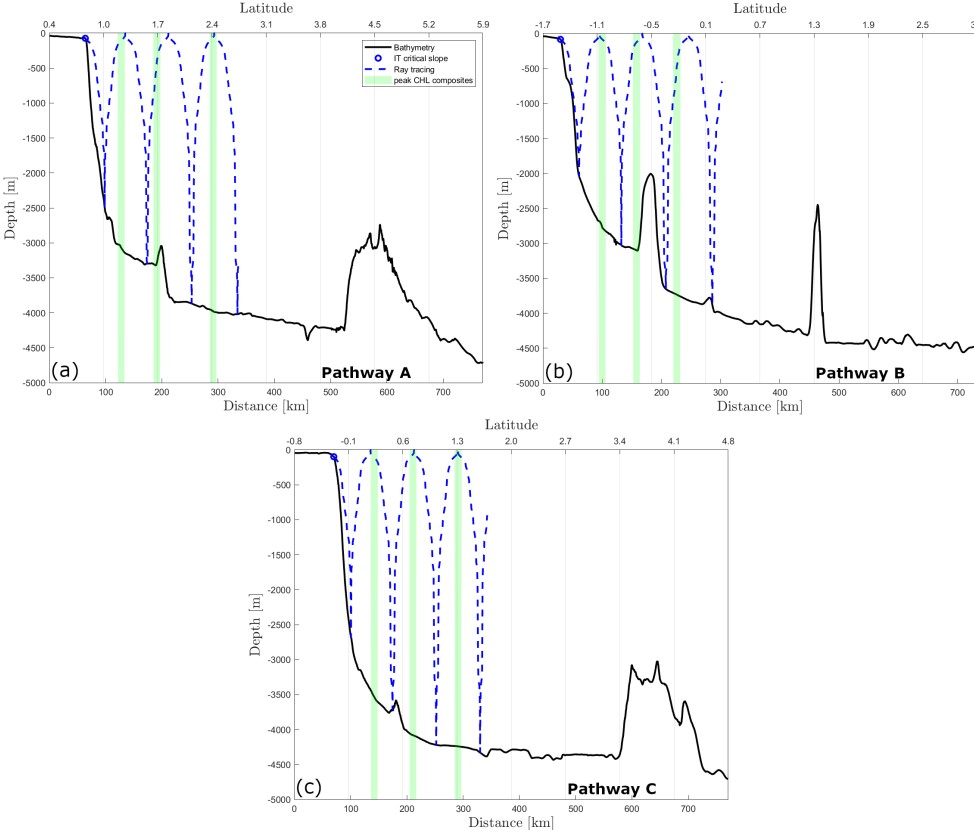

**Figure 11.** IT ray paths originating from the calculated critical slope along the IT pathways: (a) A, (b) B, and (c) C. The black line represents the bathymetry, while the green vertical bands indicates the positive peaks in the CHL composite derived from the GlobColour product.

IT patterns that shift locations throughout the year, likely diminishing the IT signal in CHL composites. For instance, CHL modulation in the showcases we presented in Section 3.1 can be up to 30% higher than the CHL 15-day average, i.e., much higher than the weaker signal in the CHL composites.

*Different processes of influences*

355    The wave patterns observed in CHL spring-neap tide composites from GlobColour and MODIS-AQUA exhibited modulations in signals filtered for both mode-1 and mode-2 internal tide (IT) wavelengths. This suggests that the patterns may arise from a combination of mode-1 and mode-2 ITs, and/or interference between these modes. However, further research is required for a more comprehensive explanation.

The spring-neap tide modulation, filtered for mode-1 and mode-2 IT wavelengths, weakens beyond the initial 300–400 km

360    offshore, aligning with observations by Assene (2024) regarding oscillations in subsurface temperature along IT pathway A, and with findings by Tchilibou et al. (2022) showing that approximately 50% of IT energy is dissipated within the first 300 km from the slope following the A path.



Results suggest that the spring-neap tide composites from the GlobColour product are more effective for capturing spatial patterns of CHL modulation by ITs, thanks to their lower proportion of missing data. While some CHL modulation patterns may be lost in the noisier MODIS-AQUA composites, likely due to high cloud coverage, these composites may provide better estimates of the potential amplitude of CHL concentration variations.

### 4.3.3 Phytoplankton response time

Our results showed that the lag between the tidal potential and the peak chlorophyll variability, indicative of maximum mixing, ranges from 1 to 3 days after the spring-neap tides. This variation depends on the CHL data source used to calculate the spring-neap tide composites. The differences could arise from the use of merged versus non-merged data, as well as variations associated with different models, including atmospheric corrections and chlorophyll retrieval algorithms. Assene (2024) showed that maximum temperature variability associated with tidal mixing occurs on average 2-3 days after spring tides in the Brazilian Equatorial Margin. Similarly, Shi et al. (2011) found that maximum turbidity and suspended matter occur with a 2-day delay from spring tide in the Yellow Sea. Another delayed impact of IT mixing is the increased nutrient availability in the euphotic zone, which can lead to an increase in biomass after a few days. The exact time lag between IT mixing and productivity depends on light availability, nutrient supply induced by the tides and the phytoplankton community and its associated growth rates (Wang et al., 2011; M'hamdi et al., 2025).

### 5 Conclusions

In this study, we used remotely sensed chlorophyll-a (CHL) concentration data to provide the first analysis of how barotropic and baroclinic tides influence the surface spatial variability of CHL concentration in the Brazilian Equatorial Margin. For the first time in this region, our findings reveal the presence of the internal tide (IT) signal in CHL data in the region, identified through an extensive time series analysis.

The findings indicate that the greatest CHL fortnightly mean Morlet wavelet power and variability in the spring-neap tide composites occurs on the shallow shelf, likely due to the mixing driven by barotropic tides. In the northwestern shelf, the turbid waters of the Amazon River plume experience sediment resuspension and mixing driven by barotropic tides during spring tides, which can limit light and hinder primary production. During neap tides, reduced mixing enhances plume stratification, limiting nutrient dispersion and increasing nutrient availability within the plume. If light availability is sufficient, phytoplankton can utilize these nutrients, contributing to the observed negative spring-neap tide CHL anomaly (-50% in GlobColour and -84% in MODIS-AQUA). In the northeastern part of the shelf, where ocean waters are lower turbid, a positive spring-neap tide CHL anomaly is found likely connected to the vertical mixing of nutrients by barotropic tides which invigorate primary production in the euphotic surface layer (GlobColour and MODIS-AQUA, respectively, 30% and 70%).

Surface signatures in CHL concentration are shown to be consistent with the influence of ITs as they propagate offshore the open ocean. Two showcases based on MODIS-AQUA imagery reveal along IT pathway A bands of CHL concentration up to 30% higher than the 15-day chlorophyll average. These bands are probably associated with the passage of IT crests.





Moreover, the IT pathways A, B, and C show a distinct wave pattern in the horizontal structure of the GlobColour spring-neap tide CHL compositions, marked by a positive spring-neap tide anomaly. The spatial positions of the positive spring-neap tide CHL anomalies align well with areas of high CHL fortnightly Wavelet power. The wave pattern appears clearer and more coherent in the Globcolour composites than those from MODIS-AQUA. In the Globcolour composites, three beams of CHL-positive anomalies are visible along the three IT pathways, reaching up to approximately 3.3%. Similarly, the MODIS-AQUA

composites display three beams of CHL-positive anomalies along IT pathway A and two beams along IT C, with the beam locations aligning well with those in the Globcolour composites but with higher CHL anomalies reaching up to 9.0%. No significant signal is observed along pathways B. GlobColour and MODIS-AQUA composites' beam signal is likely attenuated after 300-400 km likely due to dissipation of the ITs as discussed in Assene et al. (2024), which disperses the signal. Spring-neap tide composites derived from the merged GlobColour product appear to be more effective for observing the spatial patterns

of CHL modulated by ITs. In contrast, composites based on MODIS-AQUA seem to provide better estimates of the potential amplitude of CHL concentration variations.

The wave-like pattern may result from two effects: (1) tide aliasing caused by the MODIS-AQUA satellite's orbit, leading it to consistently capture IT wave crests in nearly the same location combined with deep chlorophyll maximum (DCM) modulation induced by IT passage along the thermocline as interfacial waves; and 2) ITs propagating as ray-like structures

(IT beams). Our ray-tracing analysis revealed that CHL peaks in GlobColour composites align with surface IT beams, with a similar alignment observed in MODIS-AQUA data. This alignment suggests that tidal beams may be responsible for the wave patterns in the CHL composites, enhancing primary production within their path. At present it is not possible to say which of the two mechanisms described above (tide-aliasing of the positions of the DCM within the spring-neap cycle, or enhanced CHL owing to the surfacing of IT beams) is responsible for modulations in the CHL composites. More work is needed to determine

if one of the two mechanisms is dominant, or if both mechanisms are at work.

The wave pattern in the spring-neap composite maps is likely weakened by IT interactions with background circulation features such as eddies, currents, and stratification. These interactions create incoherent IT patterns that shift throughout the year, reducing the IT signal in CHL composites.

Wave patterns in CHL spring-neap tide composites from GlobColour and MODIS-AQUA suggest contributions from mode-1

and mode-2 internal tides, but further research is needed for clarity. Results indicate a lag of 1-3 days between spring-neap tides and peak chlorophyll variability, indicative of maximum mixing. Additionally, IT mixing likely enhances nutrient availability, boosting biomass after a time lag influenced by light and phytoplankton dynamics (growth rates). ITs exhibit a more distinct fortnightly variability on surface CHL than on SST. This is likely due to differences in the water column penetration of the remote sensors and the influence of the air-sea heat fluxes, which tend to damp SST variability due to ITs (Assene et al., 2024).

The ITs might be responsible of an increase of the productivity offshore of the Amazon river as in other areas in the global ocean that still need to be estimated and the associated processes understood in order to predict their possible change due to climate change. In future work, a coupled physical-biogeochemical model will be used to quantify the productivity due to ITs and disentangle the different coupled processes at work that can explained the CHL signal associated with ITs as well as the nutrient fluxes and increase of primary production.




## Appendix A

Figure A1 presents a zoom that highlights the wavelike pattern in the horizontal structure of the spring-neap tide CHL composites, with a 1-day delay for the GlobColour product and a 2-day delay for the MODIS-AQUA images. Figure A2 shows the spring–neap tide CHL composite signal filtered for mode-1 and mode-2 IT wavelengths along a profile located outside the region influenced by ITs. As expected, the filtered signal in this area is weak.

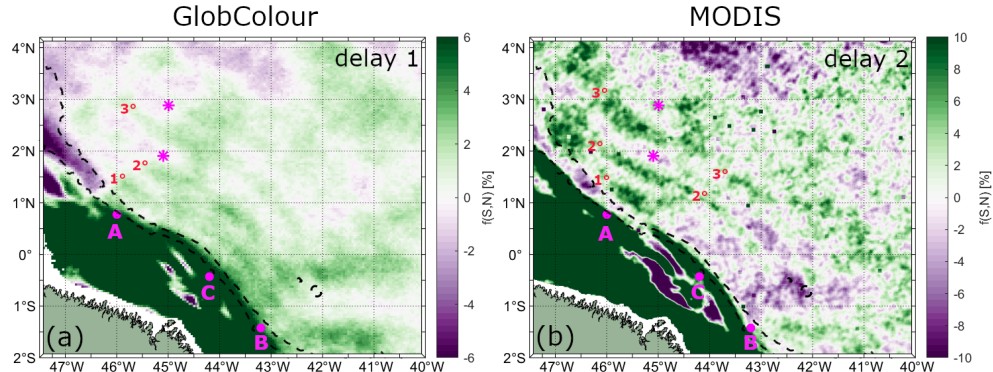

**Figure A1.** Close-up view highlighting the wave-like pattern in the horizontal structure of spring-neap tide CHL composites for (a) Glob-Colour, with a 1-day delay, and (b) MODIS-AQUA, with a 2-day delay. IT generation points, as identified by Assene et al. (2024), are indicated by magenta dots, while areas of high ISW occurrence, based on de Macedo et al. (2023), are marked with magenta stars. The numbered labels denote the first, second, and third positive peaks in the CHL composites.

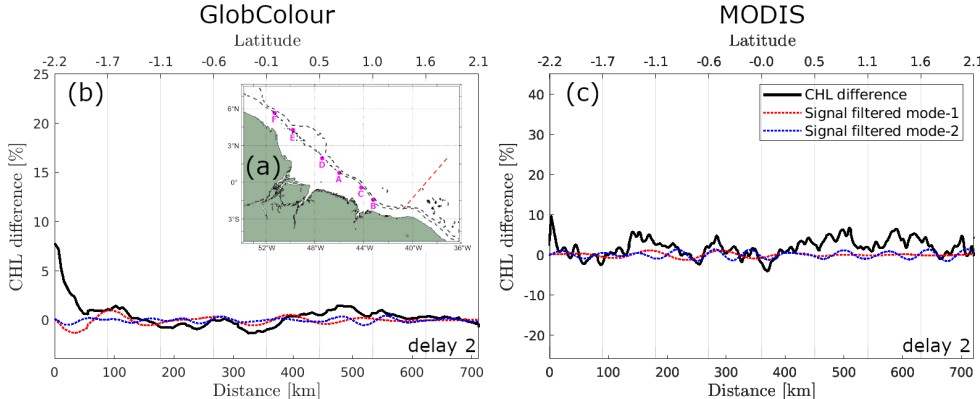

**Figure A2.** (a) Map of the study area highlighting the selected transect (red dotted line), which serves as a reference for a region unaffected by IT influence. Panel (b) shows the CHL composite profiles along this reference transect using GlobColour and (c) MODIS-AQUA. The black line represents the original signal, while the dashed red and blue lines correspond to signals filtered for mode-1 (100–150 km) and mode-2 (50–100 km) wavelengths



*Data availability.* MODIS-Terra imagery: NASA's Earth Science Data System (ESDS) (https://www.earthdata.nasa.gov/); MODIS-AQUA imagery: NASA's Ocean Color website (https://oceancolor.gsfc.nasa.gov/); CHL from GlobColour, SST from OSTIA, seawater salinity and temperature from Global Ocean Ensemble Physics Reanalysis: Copernicus Marine Service (https://marine.copernicus.eu/); Bathymetry data: GEBCO dataset (https://www.gebco.net/data$_a$nd$_p$roducts/gridded$_b$athymetry$_d$ata/).

*Author contributions.* The remote-sensing data processing was made by CRdM, with the help of MDT and TKT. Barotropic and baroclinic energy flux and dissipation computed using NEMO v4.0.2 with AMAZOMIX36 configuration was made by FA and AKL. Tide simulation was made by AKL. Analysis was performed and discussed by CRdM with the help of AKL, VV, JCBdS, JMM, ID and AMH. The paper was written with the help of all authors.

*Competing interests.* The authors declare that they have no conflict of interest.

*Acknowledgements.* The authors express their gratitude to NASA's Earth Science Data System (ESDS) for providing MODIS-Terra data, NASA Ocean Color for the MODIS-Aqua data, and Hygeos for the atmospheric correction tool, POLYMER 4.13. Additionally, thanks are extended to the SIROCCO national service for providing the COMODO-SIROCCO tools and the Copernicus Marine Service for supplying chlorophyll data from the Copernicus-GlobColour processor, SST data from OSTIA, and the Global Ocean Ensemble Physics Reanalysis data. This work is part of the "Amazomix" project (Bertrand et al., 2021)).

*Finantial support.* This work and the CDD contract from CRdM were supported by CNES funding within the framework of the APR TOSCA MIAMAZ TOSCA project (PIs Ariane Koch-Larrouy, Vincent Vantrepotte, and Isabelle Dadou). JCBdS was funded by national funds through FCT – Fundação para a Ciência e Tecnologia, I.P., in the framework of the UIDB/04683 and UIDP/04683 – Instituto de Ciências da Terra. JMM received support from the FCT, which funded this research through the projects THYOCEAN (https://doi.org/10.54499/2022.00471.CEECIND/CP1728/CT0002) and MIWAVES
(https://doi.org/10.54499/2022.01215.PTDC), as well as through projects UIDB/04423/2020, UIDP/04423/2020, and LA/P/0101 /2020.



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
