# Peer review of "Internal tide signatures on surface chlorophyll concentration in the Brazilian Equatorial Margin"

_EGUsphere, 2025_

## Author Comment (AC1)

**RESPONSE TO RC1'S COMMENTS**

**Manuscript Title:** Internal tide signatures on surface chlorophyll concentration in the Brazilian Equatorial Margin

**Manuscript ID:** EGUSPHERE-2025-2307

Journal: Ocean Science

Dear Reviewer,

We sincerely appreciate your comments and the suggestions provided. In the following pages, we address each point raised and detail the corresponding changes made to the manuscript. We hope that the revised version meets your expectations and is now suitable for publication in Ocean Science.

Sincerely,

Dr. Carina Regina de Macedo, Dr. Ariane Koch-Larrouy, Prof. José Carlos Bastos da Silva, Dr. Jorge Manuel Magalhães, Dr. Fernand Assene, Dr. Manh Duy Tran, Dr. Isabelle Dadou, Mr. Amine M'Hamdi, Dr. Trung Kien Tran, and Dr. Vincent Vantrepotte

Note: In the revised manuscript, all modifications are marked in red.

**REVIEWER 1'S COMMENTS:**

**Merit:**

The manuscript is concise and well written. I think the material would fit well in EGU Ocean Science and I anticipate it ultimately will be a nice contribution. While the figures and results are mostly convincing, I think there are a few weak points in the presentation and analysis, which I have mentioned below.

Major comments (see line-by-line comments for more details):

1. The findings are strongly based on two "showcase" days, and it is unclear whether they are necessarily reflective of what is typically observed in this region.

**Response:** The two days were chosen because they clearly displayed the patterns previously linked to ITs in other studies (da Silva et al., 2002; Muacho et al., 2014), which were also observed in other CHL images in our study area. This indicates that ITs likely influence chlorophyll concentration levels. However, at this stage, it is difficult to claim that these cases

are representative of the typical conditions in the region. Further details and explanations are provided in the discussion of the "line-by-line comments and suggestions" later in the text.

**2. The spring-neap tide composite needs to be discussed in more detail with clearer physical context.**

**Response:** The spring–neap tidal cycle composite maps capture the broader MSf (Lunisolar Synodic Fortnightly) frequency. Accordingly, we revised Section 2.4 (spring–neap tidal cycle composite) to better clarify this physical context:

lines 139-144: We developed an alternative methodology to investigate the fortnightly tidal modulation of CHL by constructing spring—neap tidal cycle composite maps. Assuming that the IT signal is coherent (i.e., coherent ITs), these composites can resolve the broader MSf frequency. However, unlike wavelet analysis or other methods commonly applied to examine the frequency content of IT-related signals (e.g., least-squares harmonic analysis; Zaron et al., 2023), our approach also preserves the sign of the spring—neap CHL variability. This enables us to determine whether tidal at MSf frequency influence drives increases or decreases in CHL concentration within the affected region. If the signal associated to ITs is coherent (i.e. coherent ITs) the composite maps will help us to extract the MSf frequency.

Because coherent internal tides are largely governed by the spring—neap cycle of the barotropic forcing, we expect our methodology to capture the associated CHL modulations at the MSf frequency. Nevertheless, as ITs propagate away from their generation sites, incoherence grows and phase coherence is gradually lost, which can partially disrupt the signal (see Section 4.3.2 for the discussion on incoherent ITs and the wave-like patterns observed in the composites).

**3. Several of the assumptions are not sufficiently justified and may be a source of uncertainty. These include a) exclusion of turbid waters when processing satellite data, b) assumption of stratification in ray tracing.**

**Response:** We agree with the reviewer that the reason and the process for excluding turbid waters were not clear in the manuscript. This decision was made because chlorophyll algorithms are particularly vulnerable to inaccuracies in highly turbid waters (Tran et al., 2023), a condition strongly influenced by the substantial sediment input from the Amazon River. The manuscript has been revised to clarify this point more explicitly. Additionally, the methodological approach has also been described in greater detail:

lines 102-109: Models based on band ratios in the visible spectrum offer reliable chlorophyll estimates for clear to moderately turbid waters but show limitations in highly turbid environments (Tran et al., 2023). Considering the presence of highly turbid waters in our study area, associated with the Amazon River plume, we applied the methodology developed by Tran et al. (2023) to exclude information from highly turbid locations. Specifically, if a given location (longitude, latitude) was classified as turbid waters (optical water types 4 and/or 5) in more than 20% of the time series, all data from that location were excluded from the analysis. This

approach helped avoid bias in the time series by removing pixels that were permanently or episodically associated with turbid environments. Overall, the excluded pixels are mainly confined to areas shallower than 100 m, where there is no evidence the ISWs discussed in this manuscript.

Regarding the assumption of gradual and continuous ocean stratification, indeed, the stratification (N) may vary along the internal tide pathways. We decided to remove the ray tracing experiments from the revised version of the manuscript. Further details on this modification are provided below in the corresponding responses.

4. In several places, discussion is qualitative even though it appears that quantitative results could be deduced from the figures. I think that including additional quantitative results in the text would strengthen the findings.

**Response:** As suggested by the reviewer, we have included additional quantitative results in the manuscript. Specifically, we calculated the coherence between the signal along each pathway and its corresponding signal filtered for mode-1 and mode-2 wavelengths.

Line-by-line comments and suggestions:

L8 – Perhaps it would be better to state that "chl was higher during neap tides", etc. rather than the "chlorophyll differences" term which could be confusing (in the abstract only).

**Response:** We agree with the reviewer's suggestion and have replaced the term "chlorophyll differences" with "chlorophyll is higher during neap (or spring) tides" throughout the abstract.

**L11 – clarify what the composite means if you mention in the abstract**

**Response:** We agree with the reviewer that the term "composite" should be clarified if mentioned in the abstract. However, we have decided to remove this term from the abstract to improve clarity.

**L19 - Clarify that this means chl max/min values occur 1-3 days after the tidal signal**

**Response:** To clarify the meaning of CHL max/min values occurring 1–3 days after the tidal signal, we revised the text as follows:

lines 11-12: A 1-3 day lag between higher CHL variability and tidal potential may indicate delayed nutrient mixing post-spring-neap tides.

L32 – I would recommend to add a bit more detail on how season conditions vary and how this modulates ISWs

**Response:** Following the reviewer's suggestion, we provided in the Introduction section additional details on the impact of seasonal variations on ISW modulation.

lines 26-28: During the months from August to December, the combined effect of stronger background currents and a deeper, less stratified pycnocline leads to greater variability and longer mean ISW wavelengths (Magalhães et al., 2016; de Macedo et al., 2023).

**L56 – "remote sensing"?**

**Response:** The term 'remote sensors' has been replaced with 'remote sensing', as suggested by the reviewer.

L71 – Do you mean "intra-annual" (or if not, please explain further)? As written it implies variability over many years, and I don't understand how that would be impacted by tides.

**Response:** The paper by Sharples (2008) focuses on numerical models used to simulate the spring—neap tidal cycle and its impact on annual primary production rates and vertical carbon fluxes near shelf sea fronts. According to the author, when tidal currents are weak, the model predicts inter-annual variability in the timing of the spring bloom of up to 3 days. In contrast, under strong tidal currents, the predicted variability in bloom increases to 8 days. We have rewritten the paragraph as follows:

lines 62-64: According to Sharples (2008), the fortnightly modulation of tidal mixing affects time and magnitude of primary production in stratified shelf areas; changes in the timing of the spring-neap cycle could account for up to 10% of the total inter-annual variability of bloom timing in regions with weak tidal regimes, and up to 25% in areas with stronger tidal currents.

L94 – Please explain why those two days were chosen. Were they objectively chosen? Are they representative of commonly observed conditions? I think it's still probably ok if not, but in that case it should be more clear if these may not be typical of what's usually observed.

**Response:** The two days were selected because they clearly exhibited the patterns discussed, which we also observed in other CHL images (see additional examples below). This suggests that ITs (and also ISWs, although beyond the scope of this manuscript) likely influence chlorophyll concentration levels in the study area, supporting the extrapolation to a 17-year time series of chlorophyll images. Nevertheless, at this stage, it is difficult to state that these cases are representative of the typical conditions in the study area. Persistent cloud cover, particularly from January to July, limits access to both MODIS-Aqua and MODIS-Terra imagery, preventing us from assessing whether such conditions also occur during months of higher cloud coverage, for which we have no significant examples. Moreover, there are cases where data are available and ISW signatures are present, and no corresponding signature in chlorophyll can be detected,

indicating that this phenomenon is more complex than explored in this manuscript. Still, we demonstrated the detectable influence of ITs on the surface chlorophyll concentration over a 17-year time series, which provides valuable evidence and opens avenues for future research to address the gaps not covered in this manuscript.

As suggested by the reviewer, we have added this concern to the manuscript by writing the following sentence:

lines 168-172: We selected two illustrative cases to demonstrate how the passage of ITs can influence chlorophyll concentration, thereby justifying the subsequent analysis that extrapolates this effect to a 17-year time series. However, it is important to note that, although these cases exhibit chlorophyll modulation patterns previously associated with ITs in other studies (da Silva et al., 2002; Muacho et al., 2014), they are not necessarily representative of the typical conditions in our study area, particularly given the lack of data during months with high cloud coverage (from January to July).

FIGURE 1:

Figure 2

L117 – I am concerned that such an exclusion will be seasonally dependent; i.e., discharge of the Amazon, and presumably turbidity as well, will vary seasonally. Thus, the excluded data will be disproportionally from a certain time of year. Please explain/quantify whether this is the case, and how this choice may have influenced later analysis and results.

Response: We agree with the reviewer that turbidity is seasonally dependent, and therefore a single location (especially in areas influenced by the Amazon River plume) may change its water type classification throughout the year. To avoid the risk of seasonally biased pixel exclusions (likely concentrated during the rainy season), we decided to completely remove from our dataset the information related to locations classified as optical water type 4 and/or 5 in more than 20% of the time series. This choice was made considering that the chlorophyll algorithm is highly prone to failure under high turbidity conditions (Tran et al., 2023), especially taking into account the large sediment load carried by the Amazon River. The manuscript has been revised to provide a clearer explanation of this concept. While this exclusion resulted in the loss of information on the impact of tides in certain key areas, such as the Amazon River mouth and adjacent coastal regions, it ensured greater confidence in the obtained results. The revised paragraph in the manuscript is presented below:

lines 102-109: Models based on band ratios in the visible spectrum offer reliable chlorophyll estimates for clear to moderately turbid waters but show limitations in highly turbid environments (Tran et al., 2023). Considering the presence of highly turbid waters in our study area, associated with the Amazon River plume, we applied the methodology developed by Tran et al. (2023) to exclude information from highly turbid locations. Specifically, if a given location (longitude, latitude) was classified as turbid waters (optical water types 4 and/or 5) in more than

20% of the time series, all data from that location were excluded from the analysis. This approach helped avoid bias in the time series by removing pixels that were permanently or episodically associated with turbid environments. Overall, the excluded pixels are mainly confined to areas shallower than 100 m, where there is no evidence the ISWs discussed in this manuscript.

L149 – I don't understand exactly what S and N are. Are they the concentration of CHL corresponding to the center of the spring/neap time or something else?

**Response:** S and N represent the CHL concentrations at the maximum and minimum tidal range elevation, respectively, spring and neap tides. To clarify this in the manuscript, we have revised the phrase as follows:

lines 148-149: Let S and N denote the CHL concentrations at the maximum and minimum tidal range elevation, respectively, spring and neap tides during the time series period (please, see Figure 2).

L155 – What do you mean by "composite maps"? I don't see any spatial term in Equation 3. From the equation it looks to me like the lagged dependence of chl on tidal cycles. Is this applied to different points in space? I am probably missing something here; please explain this in more detail.

Reading on, I can see the figures of this metric. I still think more explanation here would be helpful.

**Response:** The composite maps correspond to the weighted spring-minus-neap CHL difference calculated for each pixel p. In practice, this means that the metric is computed independently at every pixel location, thus producing a spatially distributed (map-like) representation.

In addition to the composite map centered on the spring and neap tide events, we also computed alternative composites by introducing temporal lags. Specifically, we considered delayed versions of spring and neap tides, defined as  $S=\{S1+delay, S2+delay,...,Sn+delay\}$  and  $N=\{N1+delay, N2+delay,...,Nn+delay\}$ , whith  $delay=\{0,1,2,3,4,5\}$  days.

To improve clarity, we have rewritten Section 2.4 *Spring—neap tidal cycle composite* to explicitly describe both the spatial nature of the composite maps (pixel-wise calculation) and the use of lagged versions of the spring—neap cycle:

lines 148-163: Let S and N denote the CHL concentrations at the centers of the spring and neap tidal cycles, respectively, during the time series period (please, see Figure 2). The composite maps, i.e., the weighted spring-minus-neap CHL difference for pixel p,  $f_p(S,N)$ , averaged over all cycles, can be calculated as follows:

$$f_p(S,N) = \frac{1}{n} \sum_{i=1}^n \frac{\left(\frac{3}{4}S_{p,i}^{(w)} + \frac{1}{4}S_{p,i+1}^{(w)}\right) - \left(\frac{1}{4}N_{p,i}^{(w)} + \frac{3}{4}N_{p,i+1}^{(w)}\right)}{CHL_i^{(w)}}$$
(3)

where

$$S_{p,i}^{(w)} = \frac{1}{3} \sum_{j=-1}^{+1} S_{p,t_{S,i}+j} \quad , \quad N_{p,i}^{(w)} = \frac{1}{3} \sum_{j=-1}^{+1} N_{p,t_{N,i}+j}$$
(4)

 $S_{p,ts,i+j}$  ( $N_{p,t_{N,i}+j}$ ) is the CHL value at pixel p on day  $t_{S,i}+j$  ( $t_{N,i}+j$ ), where  $t_{S,i}$  ( $t_{N,i}$ ) is the nominal day of the i-th spring (neap) tide event, i.e., j=0, and  $j \in \{-1,0,+1\}$  days from spring tides.  $CHL_i^{(w)}$  represents the mean chl concentration used to normalize each spring-minus-neap CHL difference in the series. Specifically, for each index i,  $CHL_i^{(w)}$  is computed as the average of all chlorophyll-a values included in the calculation of the weighted difference, considering both the spring and neap tide observations within the local 3-pixel window. This normalization allows each difference to be expressed relative to the local chl magnitude, ensuring comparability across the time series. In addition to the composite map centered on the spring and neap tide events, we also computed alternative composites by introducing temporal lags. This approach accounts for the phase lag between the maximum tidal potential and the maximum tidal elevation (the so-called age of the tide), when mixing intensity typically peaks. Specifically, we considered delayed versions of spring and neap tides, defined as  $S=\{S1+delay, S2+delay,...,Sn+delay\}$  and  $N=\{N1+delay, N2+delay,...,Nn+delay\}$ , where  $delay=\{0,1,2,3,4,5\}$  days.

L163 – I do not think these stratification assumptions are valid in the region of study. Because of the Amazon plume, I suspect that the profile of N is highly variable and not gradual nor continuously varying (you seem to also mention this at L173). Are there any references that show how much N varies? I think some measure of uncertainty should be included and quantified.

**Response:** We appreciate the reviewer's observation. Indeed, the stratification (N) may vary along the internal tide pathways. However, since the wave-like pattern occurs within approximately 400 km from the IT generation sites—i.e., in a region located south of the main extent of the Amazon plume—this variability is more likely related to the regional current regime and mesoscale instabilities, such as eddies associated with the NECC, than to the plume's influence. Nevertheless, acknowledging that N may not vary gradually along the propagation

pathways, we decided to remove the ray tracing experiments from the revised version of the manuscript. This modification is reflected in the Results section as follows:

lines 340-349: The wave-like pattern could potentially arise from two different mechanisms, although these remain hypothetical: (1) Tidal aliasing combined with the modulation of the DCM induced by the passage of interfacial ITs. Interfacial IT waves can lift the DCM above the light penetration depth, making it detectable by remote sensing instruments (da Silva et al., 2002; Muacho et al., 2014; Kim et al., 2018; M'Hamdi et al., 2025). As for tidal aliasing, this would imply that MODIS-Aqua repeatedly observes IT crests in nearly the same location, given that both IT generation and the satellite orbit are synchronized with the M2 tidal constituent. However, this explanation remains speculative and requires further investigation, especially considering that two semidiurnal tidal cycles are merged and averaged in the one-day GlobColour product. (2) Internal tide beams, which could potentially reduce subsurface light limitation for primary production and enhance nutrient fluxes that further stimulate biological activity (Althaus et al., 2003; Jacobsen et al., 2023; Kouogang et al., 2024). Additional research is needed to better understand the origin of the observed wave-like pattern.

**Fig 2 - I would recommend to include in the caption or figure what "difference" specifically refers to.**

**Response:** We have specified in the figure caption what the term "difference" refers to. The revised version is provided below.

[...] (c) CHL relative difference (%) between the CHL on the day of ISW occurrence and the 15-day mean CHL centered on that day (see Equation 1) from GlobColour product [...].

Fig 2 a/b (and other figures) – I would suggest to change the color scale. Rainbow color scales are not perceptually uniform, and thus the magnitude of features such as fronts may be enhanced/biased by the scale. I would recommend to use "cmocean" or a similar perceptually uniform scale.

**Response:** As suggested by the reviewer, we modified the color scale of Figures 2(a) and 2(b) to the cmocean scale. The updated figures are shown below.

**Figure 3.** Illustrative case I showing the influence of ITs on CHL concentration on September 28, 2007. CHL concentration data is shown from (a) MODIS-Aqua and (b) Globcolour product. (c) CHL relative difference (%) between the CHL on the day of ISW occurrence and the 15-day mean CHL centered on that day (see, Equation 1) from GlobColour product. (d) ISW signatures observed in the MODIS-Terra image. The red dashed line and magenta dots indicate the IT pathway and generation points, respectively, based on Assene et al. (2024). Black dashed lines mark the ISW signatures visible in the MODIS-Terra image.

**Figure 4.** Illustrative case II showing the influence of ITs on CHL concentration on October 12, 2018. CHL concentration data is shown from (a) MODIS-Aqua and (b) Globcolour product. (c) CHL relative difference (%) between the CHL on the day of ISW occurrence and the 15-day mean CHL centered on that day (see, Equation 1) from GlobColour product. (d) ISW signatures observed in the MODIS-Terra image. The red dashed line and magenta dots indicate the IT pathway and generation points, respectively, based on Assene et al. (2024). Black dashed lines mark the ISW signatures visible in the MODIS-Terra image.

L213 – For clarity, the point here is that the mode 2 filtered signal is strongly correlated with the chlorophyll difference, correct? It might be helpful to quantify this correlation in some way.

**Response:** Following the reviewer's suggestion, a few comments below, we quantified the correlation by calculating the coherence between the signal filtered to mode-1 and mode-2 IT wavelengths and the chlorophyll difference.

L215/Fig 5 – There needs to be a physical explanation of what the result of the wavelet analysis is showing. It is just the spectral energy at a period of 15 days so an indicator of regions where spring/neap tides are strongest, correct? Or maybe I am missing a bit of it?

**Response:** The spectral energy at a period of  $\sim$ 14.7 days (the fortnightly signal) indicates variability in the time series associated with fortnightly oscillations. To clarify this point, we have added further explanation in Section 2.3 (Wavelet analysis):

lines 127-130: In regions where M2 and S2 tidal constituents dominate, their nonlinear interaction generates the MSf (Lunisolar Synodic Fortnightly) oscillation, with a period of approximately 14.7 days. The MSf corresponds to the neap—spring tidal cycle, a phenomenon of great importance in tidal dynamics and a major physical factor influencing coastal and marine environments. Wavelet analysis was therefore applied to identify and quantify this fortnightly variability in the CHL and SST time series.

**L252 – It would be helpful to quantify the coherence of the signals for each of the pathways. I agree with the findings visually, but having a quantitative comparison would make your argument stronger.**

**Response:** We thank the reviewer for the suggestion. To provide a quantitative assessment of the contribution of each internal tide mode, we computed the mean spectral coherence between the original CHL signal along each transect and the band-pass filtered components corresponding to mode-1 (100–150 km) and mode-2 (50–100 km) wavelengths (Rabiner and Gold, 1975; Kay, 1988; Welch, 1967). This allows us to quantify the relative contribution of each mode to the observed CHL variability for each pathway. The results are now included in Table 2 (for GlobColour and MODIS-Aqua datasets, respectively), showing the coherence values. We also included this results in the Results and Discussion Sections, as follows.

lines 207-211: For each transect, the mean spectral coherence (Rabiner and Gold, 1975; Kay, 1988; Welch, 1967) between the original CHL relative difference signal and the band-pass filter components (mode-1 and mode-2) was computed to quantify the relative contribution of each internal tide mode. For illustrative case I, the mean spectral coherence values for mode-1 and mode-2 are 0.11 and 0.23, respectively; for case II, they are 0.17 and 0.26, respectively.

lines 249-251: The mean spectral coherence between spring-neap tidal cycle signal and the band-pass filter component is higher for mode-2 than mode-1, considering all delays (see Table 2).

lines 276-277: In both pathways A and C, the signal filtered for mode-2 IT wavelength is more coherent than the signal filtered for mode-1 (see Table 2).

We also added this information in the "Discussion" and "Summary and conclusions" sections:

lines 359-361: The mean spectral coherence between spring-neap tidal cycle signal and the band-pass filter component is higher for mode-2 than mode-1, considering both GlobColour and MODIS-Aqua data.

lines 406-407: Wave patterns in CHL spring-neap tidal cycle composites from GlobColour and MODIS-Aqua suggest contributions from mode-1 and mode-2 ITs, with mode-2 components having higher spectral coherence with the original signal.

Table 2. Mean spectral coherence for modes 1 and 2 along IT pathways (A, B, C) for GlobColour and MODIS-Aqua dataset at different delays.

| Delay | IT pathway | Coherence mode-1 |            | Coherence mode-2 |            |
|-------|------------|------------------|------------|------------------|------------|
|       |            | GlobColour       | MODIS-Aqua | GlobColour       | MODIS-Aqua |
| 0     | A          | 0.29             | 0.28       | 0.83             | 0.65       |
|       | В          | 0.01             | 0.39       | 0.72             | 0.82       |
|       | С          | 0.34             | 0.32       | 0.63             | 0.72       |
| 1     | A          | 0.50             | 0.51       | 0.82             | 0.80       |
|       | В          | 0.03             | 0.13       | 0.71             | 0.80       |
|       | С          | 0.46             | 0.18       | 0.59             | 0.74       |
| 2     | A          | 0.58             | 0.34       | 0.82             | 0.80       |
|       | В          | 0.06             | 0.18       | 0.52             | 0.66       |
|       | С          | 0.52             | 0.24       | 0.60             | 0.69       |
| 3     | A          | 0.49             | 0.40       | 0.81             | 0.78       |
|       | В          | 0.05             | 0.09       | 0.57             | 0.38       |
|       | С          | 0.54             | 0.39       | 0.59             | 0.75       |
| 4     | A          | 0.37             | 0.33       | 0.78             | 0.61       |
|       | В          | 0.05             | 0.19       | 0.68             | 0.43       |
|       | С          | 0.49             | 0.17       | 0.71             | 0.66       |
| 5     | A          | 0.34             | 0.56       | 0.73             | 0.41       |
|       | В          | 0.07             | 0.31       | 0.74             | 0.60       |
|       | С          | 0.38             | 0.17       | 0.75             | 0.63       |

Fig 7 – This figure only shows lags up to 3 days. I would include days 4 and 5; otherwise it is unclear that there is not an even stronger signal with more time lag.

**Response:** As suggested by the reviewer, we have extended Figures 8 and 10 to include lags up to 5 days. Please, see the Figures below.

Figure 8. Profiles along the A, B, and C pathways (columns) of thespring-neap tidal cycle CHL composite using Globcolour, considering delays of 0-5 days. black and dashed red and blue lines represent, respectively, the original signal and the signal filtered for mode-1 (100-150 km) and mode-2 (50-100 km) wavelengths.

Figure 10. Profiles along the A, B, and C pathways (columns) of the spring-neap tidal cycle CHL composite using MODIS-Aqua, considering delays of 0-5 days. black and dashed red and blue lines represent, respectively, the original signal and the signal filtered for mode-1 (100-150 km) and mode-2 (50-100 km) wavelengths.

L321 – I'm still not quite convinced that aliasing does not significantly impact these results; I suspect there will be regional differences from the North Sea to the area of study. Are there other references that have looked at this in similar conditions (and found that the bias was between flood/ebb and not high/low)? In any case, I think more explanation beyond citing a single study would be helpful.

**Response:** We acknowledge the reviewer's concern regarding potential tidal aliasing effects related to low-high tide. It is worth noting that satellite overpasses occur at approximately the

same local time each day, which minimizes potential impacts on chlorophyll related to light-dependent processes. In addition, in offshore regions, the influence of barotropic tides is considerably weaker than in coastal areas, reducing the likelihood of strong high-low tidal aliasing effects. Some studies have suggested that physical resuspension of particles and phytoplankton during periods of strong currents between high and low tides (~6 h tidal periodicity) can modulate chlorophyll concentrations (Yin and Harrison, 2000; Blauw et al., 2012; Blauw et al., 2018). Nevertheless, to the best of our knowledge, no study has specifically investigated chlorophyll variations between high and low tides in our study region. We have incorporated this discussion into the manuscript as follows:

lines 327-329: Nevertheless, to the best of our knowledge, we could not find any study that has specifically addressed chlorophyll variations between high and low tides in our study area.

As an additional robustness is that we compared spring—neap tidal cycle composites from GlobColour (an ensemble product derived from multiple sensors and therefore less prone to tidal aliasing) with those from MODIS-Aqua. Both products show very similar patterns, which strengthens our confidence that high-low tidal aliasing does not significantly bias our results.

L350 – These factors would be expected vary seasonally. I would recommend to discuss in a bit more detail what conditions were present in the two showcase times. Regarding eddies and currents this could be verified with SSH.

**Response:** Thank you for your valuable suggestion. We agree that seasonal variability may influence the observed processes. To address this, we included a more detailed description of the oceanic conditions during the two showcase periods, based on sea surface height anomalies (SLA) and derived geostrophic currents from the Copernicus Marine Service data. These analyses confirm that distinct mesoscale circulation patterns were present in each case, consistent with the seasonal dynamics of the region. However, in both illustrative cases, the ISW location remains outside and to the south of the region directly influenced by the mesoscale processes. This discussion has been added to subsection 3.1, *IT Illustrative Case Analysis* (see below).

lines 172-174: Furthermore, an examination of the background circulation and sea level anomaly (see Figure A1) for the days corresponding to the illustrative cases reveals that eddy activity is more intense during illustrative case I. However, in both periods, the ISW location remains outside and to the south of the region directly influenced by the mesoscale processes.

Figure A1. SLA (color shading) and surface geostrophic currents (arrows) for the two illustrative periods: (a) case I and (b) case II.

**L355-358 – The figures showed greater coherence with the mode 2 tides. Would that suggest those are more responsible?**

**Resposta:** We thank the reviewer for this comment. The higher coherence observed between CHL variability and mode-2 internal tides does not necessarily imply that mode-2 tides are the main drivers of the observed CHL variations. But it likely indicates that CHL is more sensitive to the physical processes typically associated with higher internal tide modes. Higher modes are more closely linked to enhanced vertical shear and mixing, which can promote nutrient entrainment into the euphotic layer and, consequently, variations in chlorophyll concentration. As discussed in previous studies, the redistribution of low-mode energy flux to higher modes through interactions with the background circulation provides an important mechanism for driving mixing away from internal tide generation sites. Scattering to higher modes allows for greater vertical propagation and energy dissipation, contributing significantly to deep-ocean mixing (e.g., Kerry et al., 2014). Therefore, the stronger coherence with mode-2 internal tides likely reflects the influence of the enhanced vertical mixing and instability commonly associated with higher-mode processes. However, further investigation would be required to confirm whether this mechanism is indeed dominant in our study region, as the relative contribution of each mode may vary spatially and seasonally depending on local stratification and bathymetric features.

**All this information was added in the Discussion section as follow:**

lines 359-366: The mean spectral coherence between spring-neap tidal cycle signal and the band-pass filter component is higher for mode-2 than mode-1, considering both GlobColour and MODIS-Aqua data. It can indicate that CHL is more sensitive to the physical processes typically associated with higher internal-tide modes. As discussed in previous studies, the redistribution of low-mode energy flux to higher modes through interactions with the background circulation provides an important mechanism for driving mixing away from internal-tide generation sites; scattering to higher modes allows for greater vertical propagation and energy dissipation (Kerry

et al., 2014; Dunphy & Lamb, 2014; Savva et al., 2018; Tuerena et al., 2019; Lahaye et al., 2020; Li et al., 2023).

L384 (and in other places) – This study does not directly show that there is mixing due to tides. It is inferred based on surface measurements. This should be explicitly clear somewhere.

**Response:** We agree with the reviewer that it is important to state explicitly that our study is based on surface measurements and does not directly measure mixing induced by tides. To clarify this point, first of all we have excluded this (and all the discussion) from the Summary and conclusions section. We have revised the text (in Discussion section) as follows:

lines 289-291: The findings indicate that the highest fortnightly mean Morlet wavelet power of CHL occurs on the shallow shelf, where barotropic tides dissipate through bottom friction (see Figure 6-(c)). Based on the surface measurements, we infer that this dissipation induces mixing.

In line with this point, we have also revised the text in other sections.

**L389 – "less turbid"**

**Response:** As suggested by the reviewer, the term 'lower turbid' has been replaced with 'less turbid'.

**L395 – Might be helpful to refer to these pathways by longitude here as well**

**Response:** We appreciate the reviewer's suggestion. However, adding a third axis to the figures would make them visually cluttered and reduce their readability. We chose to present the pathways in terms of latitude because the variability along latitude is much greater than along longitude, which allows the features of interest—particularly the positive spring—neap tide anomalies—to be more clearly identified.

**L425 – Run-on sentence. Please reword.**

**Response:** As suggested, we have reworded the sentence as follows:

lines 410-412: ITs might be responsible for an increase in productivity offshore of the Amazon River, as in other areas of the global ocean. However, these effects still need to be better quantified, and the associated processes understood to predict their potential changes due to climate change.

L428 – Do not directly state what your future work is. I think a reworded version of the previous sentence could be used to motivate it without saying it directly.

**Response:** Following the reviewer's suggestion, we reworded the sentence as follows:

lines 412-414: In the future, a coupled physical—biogeochemical model could be employed to quantify the productivity associated with ITs off the Amazon shelf, helping to disentangle the various processes that may explain the CHL signal and to estimate nutrient fluxes.

We thank the reviewer again for the valuable comments, which have significantly contributed to improving the clarity and accuracy of our manuscript.

**REFERENCES**

Blauw, A. N., Beninca, E., Laane, R. W., Greenwood, N., & Huisman, J. (2012). Dancing with the tides: fluctuations of coastal phytoplankton orchestrated by different oscillatory modes of the tidal cycle. PLoS One, 7(11), e49319.

Blauw, A. N., Benincà, E., Laane, R. W., Greenwood, N., & Huisman, J. (2018). Predictability and environmental drivers of chlorophyll fluctuations vary across different time scales and regions of the North Sea. Progress in Oceanography, 161, 1-18.

da Silva, J. C. B., New, A. L., Srokosz, M. A., and Smyth, T. J.: On the observability of internal tidal waves in remotely-sensed ocean colour data, Geophysical Research Letters, 29, 10–1, 2002.

Kay, S. M.: Modern Spectral Estimation: Theory and Application, Prentice-Hall, Englewood Cliffs, NJ, 1988.

Kerry, C. G., Powell, B. S., and Carter, G. S.: The impact of subtidal circulation on internal tide generation and propagation in the Philippine Sea, Journal of Physical Oceanography, 44, 1386–1405, https://doi.org/10.1175/JPO-D-13-0142.1, 2014.

Muacho, S., da Silva, J., Brotas, V., Oliveira, P., and Magalhaes, J.: Chlorophyll enhancement in the central region of the Bay of Biscay as a result of internal tidal wave interaction, Journal of Marine Systems, 136, 22–30, 2014.

Rabiner, L. R. and Gold, B.: Theory and Application of Digital Signal Processing, Prentice-Hall, Englewood Cliffs, NJ, 1975.

Tran, M. D., Vantrepotte, V., Loisel, H., Oliveira, E. N., Tran, K. T., Jorge, D., Mériaux, X., and Paranhos, R.: Band ratios combination for estimating chlorophyll-a from sentinel-2 and sentinel-3 in coastal waters, Remote Sensing, 15, 1653, 2023.

Welch, P. D.: The Use of Fast Fourier Transform for the Estimation of Power Spectra: A Method Based on Time Averaging Over Short, Modified Periodograms, IEEE Transactions on Audio and Electroacoustics, AU-15, 70–73, <a href="https://doi.org/10.1109/TAU.1967.1161901">https://doi.org/10.1109/TAU.1967.1161901</a>, 1967.

Yin, K., & Harrison, P. J. (2000). Influences of flood and ebb tides on nutrient fluxes and chlorophyll on an intertidal flat. Marine Ecology Progress Series, 196, 75-85.

---

## Author Comment (AC2)

**RESPONSE TO RC1'S COMMENTS**

**Manuscript Title:** Internal tide signatures on surface chlorophyll concentration in the Brazilian Equatorial Margin

Manuscript ID: EGUSPHERE-2025-2307

Journal: Ocean Science

Dear Reviewer,

We sincerely appreciate your valuable and constructive feedback. Each of your suggestions has been thoroughly evaluated, and the corresponding changes have been incorporated into the manuscript. A detailed explanation of these revisions is provided in the following pages.

Sincerely,

Dr. Carina Regina de Macedo, Dr. Ariane Koch-Larrouy, Prof. José Carlos Bastos da Silva, Dr. Jorge Manuel Magalhães, Dr. Fernand Assene, Dr. Manh Duy Tran, Dr. Isabelle Dadou, Mr. Amine M'Hamdi, Dr. Trung Kien Tran, and Dr. Vincent Vantrepotte

Note: In the revised manuscript, all modifications are marked in red.

**REVIEWER 1'S COMMENTS:**

The manuscript (MS) makes a comprehensive and detailed analysis to connect wave-like signal in chlrophyll-a satelitte images to internal tide dynamics off the Amazon continental shelf. Although the title and abstract suggest a focus on baroclinic tides, the MS also presents ideas connecting observed patterns to barotropic tides.

**Response:** We agree with the reviewer that the original manuscript title suggested a focus on baroclinic tides. Therefore, the title has been revised to also incorporate barotropic tides:

Tide signatures on surface chlorophyll concentration in the Brazilian Equatorial Margin

The results are interesting and certainly relevant, but the MS requires extensive reorganization for the main message to come through clearly. The knowledge gap is not well-defined, and in several parts of the document, the arguments are largely speculative and not sufficiently supported by evidence. I recommend major revision. If the speculative aspects are clarified, supported by evidence, or removed, and the MS is restructured, it could make a strong contribution to the literature.

Response: Following the reviewer's suggestion, we revised the Introduction section to more clearly define the knowledge gap. Furthermore, we restructured both the Introduction and

Conclusion sections to better meet the reviewer's expectations. All the speculative parts have been removed from the conclusions. It was retained only in the discussion section, where its speculative nature is explicitly highlighted.

**MAJOR CONCERNS**

My major concerns relate primarily to structure and terminology:

**1. Structure of Introduction and Conclusion:**

The introduction reads almost like a discussion. Although the last paragraph mentions the topic of the MS, the knowledge gap or main question is unclear.

**Response:** The Introduction section has been revised to address the reviewer's expectations.

The knowledge gap is now explicitly stated in the final paragraph of the Introduction. The revised text is provided below:

lines 75-79: Despite the advances made so far, significant gaps remain concerning the impact and spatial extent of tides (particularly baroclinic tides) on CHL concentration and SST in the BEM. In this study, we investigate the fortnightly signal in remote sensing surface CHL concentration, accounting for various delays related to astronomical tidal forcing. To our knowledge, this is the first time that the influence of baroclinic tides on CHL concentration has been demonstrated in the BEM using a long-term remote sensing time series.

Similarly, the conclusion reads like a dicussion and is repetitive compared to Section 4.3. I sugget renaming it Summary and Conclusions.

**Response:** Following the suggestion of the reviewer, the Section "5. Conclusions" was renamed to "5. Summary and conclusions". In addition, all discussion elements were removed from the "Summary and Conclusions" section, which has been rewritten accordingly.

**2. Terminology:**

The use of showcase seems inappropriate in this context. Consider using case study or illustrative case.

**Response:** We agree with the reviewer's comment and have replaced "showcase" with "illustrative case.

Sunglint should not be used as an adjective.

**Response:** We thank the reviewer for pointing this out. The manuscript has been carefully revised, and all instances where "sunglint" was previously used as an adjective have been corrected.

Although I think I understand the reason behind the two study cases, the MS should clarify why they are used. It should explicit that they are intended to demonstrate the existence of internal tides in the area, which justifies exptrapolating to a time series of chlorophyll images from Globcolour and MODIS-Aqua.

**Response:** Thank you for your comment. Following the reviewer's suggestion, we have included this information in the manuscript as follows:

lines 168-179: We selected two illustrative cases to demonstrate how the passage of ITs can influence chlorophyll concentration, thereby justifying the subsequent analysis that extrapolates this effect to a 17-year time series.

Avoid using coloquial terms like "sandwich"; I recommend "to be flanked" instead.

**Response:** The term "sandwiched" has been replaced with "flanked" throughout the manuscript.

The term "spring-neap tide" is confusing; I believe you mean "spring-neap tidal cycle". Clarify or define this early in the MS.

**Response:** Following the reviewer's suggestion, we have replaced the term "spring-neap tide" with "spring-neap tidal cycle" throughout the manuscript.

Please correct the spelling of "MODIS-Aqua" consistently throughout the MS.

**Response:** The spelling of "MODIS-Aqua" has been consistently corrected throughout the manuscript.

**SPECIFIC COMMENTS (line numbers are referenced)**

51: the availability of PAR.

**Response:** We revised the sentence following the reviewer's suggestion:

lines 58-60: However, this shelf mixing can also inhibit phytoplankton growth by promoting resuspended sediments, which reduces the photosynthetically available radiation (PAR) (Byun et al., 2007; Xing et al., 2021; Kossack et al., 2023)

53: Define interfacial ITs.

**Response:** Following the reviewer's suggestion, we have added a definition of interfacial ITs:

lines 38-39: According to da Silva et al. (2002), as interfacial ITs propagate (i.e., IT waves that propagate horizontally along density interfaces) [...]

**55: Use deep or subsurface chlorophyll maximum**

**Response:** As suggested by the reviewer, we revised the term from "depth of the chlorophyll maximum (DCM)" to "deep chlorophyll maximum (DCM)".

**61-62: "to two combined effects"**

**Response:** We have corrected the sentence according to the reviewer suggestion:

lines 50-52: M'hamdi et al. (2025) demonstrated using a Slocum G2 glider deployed off the Amazon shelf during AMAZOMIX 2021 cruise that the enhancement of the CHL concentration associated with the passage of ITs may be due to two combined effects:

**72: Indonesian Seas (there are several of them) or Indonesian Throughflow**

**Response:** We agree with the reviewer's comment and have replaced the term "*Indonesian Seas*" with "*the region of the Indonesian Throughflow*".

**117: Suggestion: more than 20% of the time series.**

**Response:** According to the reviewer's suggestion, we have replaced "more than 20% in our time series" with "more than 20% of the time series".

**129: Define what is a showcase day.**

**Response:** We have defined in the manuscript "showcase day" as follows:

lines 118-119: [...] CHL concentration on the day when ISW signatures were found (i.e., the specific day of the illustrative case) [...]

163: Is this assumption valid for this region? The freshwater inflow could have drastic effects on local stratification, likely preventing gradual and continuous profiles. How might this assumption affect your analysis? You should demonstrate the validity of your assumption specially because you are looking into a surface signature of these waves.

**Response:** We appreciate the reviewer's comment. We agree that freshwater input can strongly affect vertical stratification in the Amazon plume region. However, our study area lies south of the main extent of the plume, where the observed wave-like pattern occurs within approximately 400 km from the internal tide generation sites. In this region, the variability in stratification (N) is mainly driven by the regional current regime and mesoscale instabilities, such as eddies, rather than by freshwater inflow. Nevertheless, acknowledging that N may not vary gradually along the propagation pathways, we decided to remove the ray tracing experiments from the revised version of the manuscript.

173: N (or M – the lateral buoyancy gradient) will certainly exhibit lateral variability due to riverine input and submesoscale/mesoscale stirring.

**Response:** We appreciate the reviewer's comment. As discussed in the response to the previous comment, we acknowledge that N may not vary gradually along the propagation pathways due to lateral variability induced by mesoscale/submesoscale processes. Considering this limitation, we decided to remove the ray tracing experiments from the revised manuscript.

Figure 2: Points A and B are barely visible. The magenta font over red/orange background lacks contrast. Also, clarify which quantity is shown in panel D? In addition, having the coastline would improve visualization, orientation and highlight the relevant features. ISW signatures do not stand out, significant zooming is needed to see them.

**Response:** We have followed the reviewer's suggestion and improved the contrast at points A and D. A colorbar was added to panel D to clarify the quantities, and coastlines were included in the figures. We also made an effort to enhance the contrast of the ISW signatures. The revised figures are shown below.

**Figure 3.** Illustrative case I showing the influence of ITs on CHL concentration on September 28, 2007. CHL concentration data is shown from (a) MODIS-Aqua and (b) Globcolour product. (c) CHL relative difference (%) between the CHL on the day of ISW occurrence and the 15-day mean CHL centered on that day (see, Equation 1) from GlobColour product. (d) ISW signatures observed in the MODIS-Terra image. The red dashed line and magenta dots indicate the IT pathway and generation points, respectively, based on Assene et al. (2024). Black dashed lines mark the ISW signatures visible in the MODIS-Terra image.

Figure 4. Illustrative case II showing the influence of ITs on CHL concentration on October 12, 2018. CHL concentration data is shown from (a) MODIS-Aqua and (b) Globcolour product. (c) CHL relative difference (%) between the CHL on the day of ISW occurrence and the 15-day mean CHL centered on that day (see, Equation 1) from GlobColour product. (d) ISW signatures observed in the MODIS-Terra image. The red dashed line and magenta dots indicate the IT pathway and generation points, respectively, based on Assene et al. (2024). Black dashed lines mark the ISW signatures visible in the MODIS-Terra image.

**189, 192: References for the typical wavelengths are needed.**

**Response:** We added the reference for the typical mode-1 IT wavelength, as suggested by the reviewer. The revised text is shown below:

lines 182:183 The bands are separated by approximately 100 km, typical mode-1 IT wavelengths (150-100 km).

205: Clarify whether the average is temporal only (one value) or also considers spatial differences (2D map).

**Response:** As suggested by the reviewer, we have clarified in the text that the average refers exclusively to the temporal dimension. Specifically, we have established the link between the maps of CHL relative difference and Equation 1 as follows:

The maps of CHL relative difference (see Equation 1) are shown for illustrative cases I and II in Figures 2(c) and 3(c), respectively.

In addition, we have rewritten Equation 1 to make it explicit that only temporal differences are considered, and not spatial ones. The revised text now reads:

lines 116-125: The presence of ISW signatures in two MODIS-Terra images acquired under conditions of sunglint over the BEM is used as a proxy for IT activity. To emphasize the impact of ITs on the CHL concentration, We calculated the CHL relative difference at pixel p (in %),  $\triangle CHL_p$ , defined as the deviation between the CHL concentration on the day when ISW signatures were detected (i.e., the illustrative case day) and the 15-day mean CHL values centered on that day:

$$\Delta CHL_p(\%) = \frac{CHL_{p,0} - \overline{CHL_p}}{\overline{CHL_p}} \times 100$$
(1),

where

$$\overline{CHL}_{p} = \frac{1}{15} \sum_{i=-7}^{+7} CHL_{p,i}$$
(2)

 $CHL_{p,i}$  is the CHL concentration at pixel p on day i (with i=0 the illustrative case day, i=-7 the 7th day before, and i=+7 the 7th day after). This enables the assessment of potential changes in CHL concentrations driven by ITs, compared to the CHL 15-day average within the study area.

Figure 4: Consider adding arrows to highlight peaks.

**Response:** As suggested by the reviewer, we added arrows to highlight the peaks in the figure. The figure was originally numbered as Figure 4, but since an additional figure was included in the revised manuscript, it now corresponds to Figure 5. Please see the updated figure below.

Figure 5. CHL concentration data along IT pathway A is shown from MODIS-Aqua for illustrative cases (a) I and (b) II, and from the GlobColour product for illustrative cases (c) I and (d) II. CHL relative difference is presented for illustrative cases (e) I and (f) II.

**216: statistically significant.**

**Response:** The term "statistically significantly" was revised to "statistically significant", as suggested by the reviewer.

**249: Difference relative to what? Please clarify.**

**Response:** As pointed out by the reviewer, we have clarified the sentence as follows:

lines 247-248: The positive peaks of CHL differences during the spring—neap tidal cycle, with a 1-day delay, average around 2.1% across all IT pathways.

**337-341: Here and later on. How would the DCM be visible from remote sensing? Does IT-induced mixing bring chlorophyll from the DCM to the surface?**

**Response:** The light penetration depth determines how deep remote sensing instruments can "see" into the water column. When interfacial ITs propagate, some studies have shown that they can modulate the DCM by displacing it above or below this depth (da Silva et al., 2002; Muacho et al., 2014; Kim et al., 2018; M'Hamdi et al., 2025). Such vertical displacements may result in either enhanced or reduced chlorophyll-a concentrations detectable by remote sensing. Additionally, as shown by M'Hamdi et al. (2025) mixing events associated with ITs increase

CHL concentration in both the surface and bottom layers of the water column. This concept is further discussed in the Introduction section:

lines 38-44: According to da Silva et al. (2002), as interfacial ITs propagate, they cause vertical displacements in the pycnocline and hence displace passive phytoplankton cells within the water column. This movement shifts the deep chlorophyll maximum (DCM) either above or below the light penetration depth, resulting in, respectively, increased or decreased chlorophyll-a concentrations (hereinafter referred to as CHL) detected by remote sensing (da Silva et al., 2002; Muacho et al., 2014; Kim et al., 2018; M'Hamdi et al., 2025).

Additionally, ITs enhance the vertical mixing of nutrients into the DCM, thus supporting primary production (Sharples et al., 2007; Tuerena et al., 2019; Kossack et al., 2023; Jacobsen et al., 2023). Jacobsen et al. (2023) studied the response of primary production to IT beams based on model simulations configured for an oligotrophic system with a nutricline depth below 50 meters. They found that while subsurface light limitation is reduced for passive plankton within tidal beams, leading to higher primary production rates, the dominant effect of tidal beams on primary production is the increased nutrient supply to the euphotic zone near tidal beam generation locations. M'Hamdi et al. (2025) demonstrated using a Slocum G2 glider deployed off the Amazon shelf during AMAZOMIX 2021 cruise that the enhancement of the CHL concentration associated with the passage of ITs may be due to two combined effects: 1) the ITs modulate the DCM, causing its vertical displacement and oscillation as it rises and deepens in response to IT propagation. This movement may enhance phytoplankton's light exposure, stimulating primary production; 2) Mixing events associated with ITs increase CHL concentration in both the surface and bottom layers of the water column.

To clarify this concept in the specific part of the manuscript mentioned by the reviewer, we have rewritten it as follows:

1) Tidal aliasing combined with the modulation of the DCM induced by the passage of interfacial IT waves. Interfacial IT waves can bring DCM above the light penetration depth, allowing the remote sensing instrument to observe it. Regarding tidal aliasing, this implies that MODIS-Aqua consistently captures IT crests in nearly the same location, since both IT generation and the satellite orbit are synchronized with the M2 tidal constituent.

**379: The MS focuses on internal tide signatures in surface chlorophyll; clarify why barotropic tides are also discussed.**

**Response:** Since our manuscript was primarily focused on exploiting the fortnightly signal in remote sensing surface chlorophyll concentration, some patterns associated with barotropic tides also emerged. We considered it relevant to include a discussion on this topic, as barotropic tides are directly linked to the generation of internal tides. We appreciate the reviewer's comment, and

in response, we have revised several sections of the manuscript (including the title, abstract, and conclusion) to broaden its scope and explicitly address the role of barotropic tides in addition to internal tide signatures.

386-389: These statements are speculative. Explain more clearly how the time series analysis supports these conclusions.

**Response:** We agree with the reviewer, and all the speculative parts have been removed from the conclusions. It was retained only in the discussion section, where its speculative nature is explicitly highlighted.

389: Clarify "lower turbid"?

**Response:** We made a grammatical mistake and have corrected the term from "lower turbid" to "less turbid" in the manuscript. Thank you for pointing this out.

392-393: "as they propagate offshore the open ocean", likely a typo; please check.

**Response:** we corrected the sentence "as they propagate offshore the open ocean" to "while they propagate offshore".

402: [...] beam signal is attenuated after 300-400 km [...]

**Response:** According to the reviewer's suggestion, we revised the sentence as follows:

lines 402-403: The beam signal in the GlobColour and MODIS-Aqua composites becomes attenuated after 300–400 km.

404: more effective TO OBSERVE.

**Response:** As suggested, the term "for observing" was replaced with "to observe."

417: How is stratification a "background circulation feature"?

**Response:** We agree with the reviewer that stratification should not be considered a "background feature." Therefore, we have removed this statement from the manuscript.

421-422: This statement is clearer than Lines 386-389.

**Response:** In accordance with the reviewer's suggestion, we have removed the statement that was originally included between lines 386 and 389.

423-424: Could the difference between tracers simply be because chlorophyll is biologically reactive while temperature is a physical tracer? The current explanation is overly complicated. I do not agree with your argument.

**Response:** We appreciate the reviewer's insightful comment. We agree that the difference between tracers may also be related to their distinct nature, with chlorophyll being biologically reactive while temperature is a physical tracer. However, a detailed investigation of this question lies beyond the scope of the present study. To avoid an overly speculative explanation, we decided to remove our initial interpretation and instead cite Assene (2024), who reported that IT propagation induces a surface cooling of approximately -0.2 °C. This study also highlights that ITs enhance the net heat flux at the air—sea interface, which tends to dampen the IT-induced SST cooling.

We have removed this from the Conclusion and revised the corresponding part of the Discussion section as follows:

lines 292-298: The fortnightly signal in SST is weaker than that observed in CHL and becomes even less pronounced offshore. Using two regional simulations (with and without tides), Assene (2024) estimated sea temperature anomalies along IT pathway A and reported offshore surface anomalies of approximately  $-0.2\,^{\circ}$ C. The study also showed that ITs modulate upper ocean—atmosphere interactions by enhancing the net heat flux at the air—sea interface. This enhanced flux from the atmosphere to the ocean tends to damp the IT-induced SST cooling and contributes to restoring surface temperatures (Assene, 2024). Such restoration could explain the weaker fortnightly signal in SST compared to CHL. Nevertheless, further studies are needed to understand these differences better.

We thank the reviewer again for the valuable comments, which have significantly contributed to improving the clarity and accuracy of our manuscript.

**REFERENCE**

da Silva, J. C. B., New, A. L., Srokosz, M. A., and Smyth, T. J.: On the observability of internal tidal waves in remotely-sensed ocean colour data, Geophysical Research Letters, 29, 10–1, 2002.

Kim, H., Son, Y. B., and Jo, Y.-H.: Hourly observed internal waves by geostationary ocean color imagery in the east/Japan Sea, Journal of Atmospheric and Oceanic Technology, 35, 609–617, 2018.

M'hamdi, A., Koch-Larrouy, A., Costa da Silva, A., Dadou, I., de Macedo, C. R., Bosse, A., Vantrepotte, V., Aguedjou, H. M., Tran, T. K., Testor, P., Mortier, L., Bertrand, A., Melo, P. A. M. d. C., Lee, J., Rollnic, M., and Araujo, M.: Impact of Internal Tides on Chlorophyll-a Distribution and Primary Production off the Amazon Shelf from Glider Measurements and Satellite Observations, EGUsphere, 2025, 1–36, 2025.

Muacho, S., da Silva, J., Brotas, V., Oliveira, P., and Magalhaes, J.: Chlorophyll enhancement in the central region of the Bay of Biscay as a result of internal tidal wave interaction, Journal of Marine Systems, 136, 22–30, 2014.

---

## Author Comment (AC3)

**RESPONSE TO CC1'S COMMENTS - Mr. Longyu Huang**

**Manuscript Title:** Internal tide signatures on surface chlorophyll concentration in the Brazilian Equatorial Margin

Manuscript ID: EGUSPHERE-2025-2307

Journal: Ocean Science

Dear Mr. Longyu Huang,

Thank you very much for your thoughtful and constructive comments. We carefully considered each of your suggestions and have addressed them in detail in the following pages, along with a description of the corresponding revisions made to the manuscript.

Sincerely,

Dr. Carina Regina de Macedo, Dr. Ariane Koch-Larrouy, Prof. José Carlos Bastos da Silva, Dr. Jorge Manuel Magalhães, Dr. Fernand Assene, Dr. Manh Duy Tran, Dr. Isabelle Dadou, Mr. Amine M'Hamdi, Dr. Trung Kien Tran, and Dr. Vincent Vantrepotte

Note: In the revised manuscript, all modifications are marked in red.

**CC1'S COMMENTS:**

In this manuscript, the authors investigate the influence of tides on chlorophyll-a (CHL) variability in the Brazilian Equatorial Margin using two types of daily remotely sensed CHL data from 2005 to 2021. Although the studies of internal tides (ITs) in this region have been widely documented, the ITs contributions on the marine ecology are rarely reported. This is an interesting topic and suitable for the journal Ocean Science. Overall, the current manuscript is well organized and written, the results are present clearly and provide valuable insights into the ecological impacts of ITs. Now, I recommend a minor revision and some questions and suggestions for the authors to improve the manuscript.

**Specific comments:**

1. The introduction is well written and clear, and summarizes the relevant research on internal tides in this region. However, in the main text, the authors mention lots of elements about the bathymetry, generation sites and pathways of internal tides in the Brazilian Equatorial Margin that referred from other studies. I suggest the authors give a figure to present the overview for the study region.

**Response:** As suggested by the community commentator, we added a figure in the Introduction providing an overview of the study area (see the figure below).

**Figure 1.** Baroclinic flux over the BEM. IT generation sites are labeled from A to F along the shelf break. The black dashed line and black dots represent the bathymetric contours of -100 m and -2000 m, and the generation points, respectively, following Assene et al. (2024). The North Brazil Current (NBC) and the NECC are highlighted with thick gray arrows.

**2. In Methods 2.2, why the average period is 15 days?**

**Response:** We chose a 15-day averaging window for two main reasons. First, we aimed to use a period close to the fortnightly tidal signal (14.7 days), while keeping an odd-numbered window so that the illustrative case day remained centered. Second, we empirically verified that longer averaging windows tended to smooth out the signal of interest, leading to loss of relevant variability.

3. In Methods 2.3, the physical mechanisms of wavelet analysis should be explained and say somewhat to explain the meanings of high or low Power in Figure 5 (a-b).

**Response:** The spectral energy at a period of  $\sim$ 14.7 days (the fortnightly signal) indicates variability in the time series associated with fortnightly oscillations. To clarify this point, we have added further explanation in Section 2.3 (Wavelet analysis):

lines 127-130: In regions where M2 and S2 tidal constituents dominate, their nonlinear interaction generates the MSf (Lunisolar Synodic Fortnightly) oscillation, with a period of approximately 14.7 days. The MSf corresponds to the neap—spring tidal cycle, a phenomenon of great importance in tidal dynamics and a major physical factor influencing coastal and marine environments. Wavelet analysis was therefore applied to identify and quantify this fortnightly variability in the CHL and SST time series.

4. In Methods 2.4, the tides in single ITs generation sites are extracted for the calculation of Tide Composition f(S, N). In Fig. 6-9, there are several ITs generation sites

and pathways, such as pathway A, B and C, I wonder if the f(S, N) is computed using tides extracted in one site or all sites?

**Response:** We appreciate the reviewer's question. The f(S,N) was computed using the tidal elevation extracted at the IT generation point A. However, we also calculated the phase lag of the tidal signal between generation points A and B, and A and C, which were  $10.3^{\circ}$  ( $\approx 0.36$  h) and  $3.65^{\circ}$  ( $\approx 0.13$  h), respectively. These small phase differences indicate that the tidal phase is nearly coherent among these sites, particularly considering that our methodology uses a three-day window centered on the spring/neap tide days.

5. In Figure 2 and 3, I suggest the author added the bathymetric contours for the clarity. Besides, the locations of points A and D are hard to distinguish, the point color should be changed. What the two black dashed lines in the lower left mean? The locations of mode-1 and 2 ISWs should be labeled in Figure 2 and 3 (d).

**Response:** Following the suggestions of CC1, we enhanced the contrast of points A and D as well as the internal solitary wave signatures. The purpose of Figure 2(d) is to show the MODIS-Terra image and the appearance of the ISW signatures within it. Labeling their locations directly on the figure would make the signatures even more difficult to visualize. The black dashed lines represent bathymetric contours; however, since they were not labeled, we agree with the CC1 that their meaning was unclear. To address this, we have now labeled the bathymetric contours. The revised figures are provided below.

**Figure 3.** Illustrative case I showing the influence of ITs on CHL concentration on September 28, 2007. CHL concentration data is shown from (a) MODIS-Aqua and (b) Globcolour product. (c) CHL relative difference (%) between the CHL on the day of ISW occurrence and the 15-day mean CHL centered on that day (see, Equation 1) from GlobColour product. (d) ISW signatures observed in the MODIS-Terra image. The red dashed line and magenta dots indicate the IT pathway and generation points, respectively, based on Assene et al. (2024). Black dashed lines mark the ISW signatures visible in the MODIS-Terra image.

Figure 4. Illustrative case II showing the influence of ITs on CHL concentration on October 12, 2018. CHL concentration data is shown from (a) MODIS-Aqua and (b) Globcolour product. (c) CHL relative difference (%) between the CHL on the day of ISW occurrence and the 15-day mean CHL centered on that day (see, Equation 1) from GlobColour product. (d) ISW signatures observed in the MODIS-Terra image. The red dashed line and magenta dots indicate the IT pathway and generation points, respectively, based on Assene et al. (2024). Black dashed lines mark the ISW signatures visible in the MODIS-Terra image.

**6. Line 210: Which IT generation site should be clarified.**

**Response:** Indeed, this information was previously missing from the manuscript. It refers to the IT generation site A. We appreciate your attention to this detail.

lines 204-205: For illustrative case II, two peaks are found at 122 km (8%) and 262 km (7%), respectively, 62 km and 202 km from the IT generation site A

7. Line 213: Figure 4 show that signal filtered mode-2 IT is more closely with the original signal of CHL different. If this mean that the variation of CHL is mainly caused by mode-2 IT? I wonder why the smaller horizontal and vertical scale of mode-2 IT could induce greater variation of CHL.

**Response:** We thank the reviewer for this question. The closer correspondence between the mode-2 IT filtered signal and the original CHL difference signal does not necessarily indicate that CHL variability is mainly driven by mode-2 internal tides. But it suggests that CHL is more

sensitive to the processes typically associated with higher internal tide modes. Higher modes are more closely linked to enhanced vertical shear and mixing, which can promote nutrient entrainment into the euphotic zone and, consequently, variations in chlorophyll concentration. Previous studies have shown that the redistribution of low-mode energy flux to higher modes through interactions with the background circulation provides an important mechanism for driving mixing away from internal tide generation sites. Scattering to higher modes allows for greater vertical propagation and energy dissipation, contributing significantly to deep-ocean mixing (e.g., Kerry et al., 2014). Therefore, we think that the stronger relationship between CHL variability and the mode-2 IT signal likely reflects the greater sensitivity of biological and biogeochemical responses to the enhanced vertical mixing and instability commonly induced by higher-mode internal tides. However, further investigation would be required to confirm if this mechanism is indeed dominant in our study region, as the relative contribution of each mode may vary spatially and seasonally depending on local stratification and bathymetric conditions.

**All this information was added in the Discussion section as follow:**

lines 359-366: The mean spectral coherence between spring-neap tidal cycle signal and the band-pass filter component is higher for mode-2 than mode-1, considering both GlobColour and MODIS-Aqua data. It can indicate that CHL is more sensitive to the physical processes typically associated with higher internal-tide modes. As discussed in previous studies, the redistribution of low-mode energy flux to higher modes through interactions with the background circulation provides an important mechanism for driving mixing away from internal-tide generation sites; scattering to higher modes allows for greater vertical propagation and energy dissipation (Kerry et al., 2014; Dunphy & Lamb, 2014; Savva et al., 2018; Tuerena et al., 2019; Lahaye et al., 2020; Li et al., 2023).

**8. Figure 5: the barotropic and baroclinic energy flux (in c and d) are from NEMO by Assene et al 2024, if the time align with (a) and (b)? Besides, in Line 225, why choose S2 tidal constituent instead of M2?**

**Response:** The depth-integrated barotropic and baroclinic energy fluxes and dissipation derived from the NEMO model using the AMAZOMIX36 configuration represent mean values for the year 2015, and therefore are not time-aligned with panels (a) and (b), which represent mean values from 2005 to 2021.

The S2 component was selected for comparison with the mean Morlet wavelet power within the 14.2–15.2-day period, as the barotropic dissipation is substantially lower than that of baroclinic one in the offshore region.

**9. Line 239: where are the three peaks of positive CHL difference?**

**Response:** Figure A2-(a) provides a close-up view that highlights the wave-like pattern in the CHL composite with a 1-day delay. The text has been modified to indicate that, in Figure A2-(a),

the three peaks of positive CHL differences related to the spring—neap tidal cycle can be clearly observed:

lines 235:238: Referring to the spring-neap tidal cycle composites with a 1-day delay as a benchmark, Figure 7-(b) illustrates at least three peaks of positive CHL spring-neap tidal cycle difference (please, see in Figure A2-(a) a close-up view highlighting the wave-like pattern in the CHL composite, with 1-day delay is shown).

**10. Line 242: Figure 7?**

**Response:** Thank you for pointing that out. You are correct — the figure was initially misnumbered. At the time of your review, it should have been Figure 6; however, since an additional figure was added to the manuscript, it now corresponds to Figure 7.

11. The authors should explain why the mode-2 f is much greater than mode-1 f in Figure 9, while the values of f are equivalent in Figure 7, even though two types of CHL data are used.

Response: We thank the reviewer for this insightful comment. The stronger mode-2 IT signal observed in Figure 9 when using MODIS-Aqua data may be related to differences in spatial resolution and data processing between the two CHL products. MODIS-Aqua provides observations at a higher spatial resolution (1 km) when compared to the Globcolour product (4 km), which can better capture submesoscale features and local gradients associated with mode-2 internal tides. In contrast, the GlobColour product is a merged dataset that combines multiple satellite missions and involves interpolation and smoothing procedures. These processes tend to reduce high-frequency spatial variability, leading to a weaker representation of mode-2 signals. Additionally, differences in sensor calibration and atmospheric correction among the merged datasets may contribute to the smaller mode-2 amplitudes in GlobColour. Despite these differences, both datasets consistently indicate the presence of mode-1 and mode-2 internal tide signals along the IT pathways.

The information on spatial resolution was added to the manuscript as follows:

lines 87-88: CHL data were derived from daily Level 1A acquisitions of the Moderate Resolution Imaging Spectroradiometer (MODIS) onboard the Aqua satellite, covering the period from January 1, 2005, to December 31, 2021, at a spatial resolution of 1 km.

lines 98-99: The GlobColour product was obtained from the Copernicus Marine Service (DOI: 10.48670/moi-00281) for the period from January 1, 2005, to December 31, 2021, at a spatial resolution of 4 km.

12. The display range of the color bar should be uniform. Such as (a-f), (g-h) in Figure 6 and 8, respectively, and Figure A1 (a-b).

**Response:** As suggested by the community commentator, we standardized the color bars in Figures 6, 8, and A1. The revised figures are provided below.

Figure 7. Spring-neap tidal cycle composites of CHL using daily Globcolour product, considering delays of (a-f) 0-5 days. Spring-neap tidal cycle composite map for delay of (g) 1 and (h) 2 days, using a color bar for highlighting the shelf. Areas of high CHL fortnightly power are shown as black rectangles, IT generation points are displayed as black points, IT pathways are shown as black dashed lines according to Assene et al. (2024), and black stars represent the areas of high ISW occurrence according to de Macedo et al. (2023).

Figure 9. Spring-neap tidal cycle composites of CHL using daily MODIS-Aqua data, considering delays of (a-f) 0-5 days.. Spring-neap tidal cycle composite map for delay of (g) 1 and (h) 2 days, using a color bar for highlighting the shelf. Areas of high CHL fortnightly power are shown as black rectangles, IT generation points are displayed as black points, IT pathways are shown as black dashed lines according to Assene et al. (2024), and black stars represent the areas of high ISW occurrence according to de Macedo et al. (2023).

**Figure A2.** ZOOM of Figures 7 and 9 highlighting the wave-like pattern in the horizontal structure of spring-neap tidal cycle CHL composites for (a) GlobColour, with a 1-day delay, and (b) MODIS-Aqua, with a 2-day delay. IT generation points, as identified by Assene et al. (2024), are indicated by black dots, while areas of high ISW occurrence, based on de Macedo et al. (2023), are marked with black stars. The numbered labels denote the first, second, and third positive peaks in the CHL composites.

We thank the reviewer again for the valuable comments, which have significantly contributed to improving the clarity and accuracy of our manuscript.

**REFERENCES**

Kerry, C. G., Powell, B. S., & Carter, G. S. (2014). The impact of subtidal circulation on internal-tide-induced mixing in the Philippine Sea. Journal of Physical Oceanography, 44(12), 3209-3224.